

# Towards accurate and practical drone-based wind measurements with an ultrasonic anemometer

William Thielicke[1], Waldemar Hübert[1], and Ulrich Müller[1]

[1]OPTOLUTION Messtechnik GmbH, Gewerbestr. 18, 79539 Lörrach, Germany

**Correspondence:** W. Thielicke (Thielicke@optolution.com)

**Abstract.** Wind data collection in the atmospheric boundary layer benefits from short term wind speed measurements using unmanned aerial vehicles. Fixed and rotary wing devices with diverse anemometer technology have been used in the past to provide such data, but the accuracy still has the potential to be increased. We developed a light weight drone (weight including sensor $\leq 5\,\mathrm{kg}$) with long flight endurance ($> 45\,\mathrm{min}$) for carrying an industry standard precision sonic anemometer. Accuracy

tests have been performed with the isolated anemometer at high tilt angles in a calibration wind tunnel, with the drone flying in a large wind tunnel, and with the full system flying at different heights next to a bistatic lidar reference.

The propeller-induced flow deflects the air to some extent, but this effect is compensated effectively. Our data fusion shows no signs of crosstalk between ground speed and wind speed. When compared with the bistatic lidar in very turbulent conditions, with 10 seconds averaging interval and with the UAV constantly circling around the measurement volume of the lidar reference,

wind speed measurements have an average absolute bias of 1.9% ($0.073\,\mathrm{m\,s^{-1}}$), wind elevation average absolute bias is $0.5°$, and wind azimuth average absolute bias is $1.5°$, indicating excellent accuracy under challenging and dynamic conditions. The system was finally flown in the wake of a wind turbine, successfully measuring the spatial velocity deficit distribution during forward flight, yielding results that are in very close agreement to lidar measurements and the theoretical distribution. We believe that the results presented in this paper can provide important information for designing flying systems for precise air

speed measurements either for short duration at multiple locations (battery powered) or for long duration at a single location (power supplied via cable). UAVs that are able to accurately measure three-dimensional wind might be used as cost effective and flexible addition to measurement masts and lidar scans.

# 1  Introduction

## 1.1  Wind speed measurements

Measurements of wind characteristics are important in the environmental science of the atmospheric boundary layer (ABL). They are crucial for predictions of meteorological processes (e.g. Lauer and Fengler, 2017), optimization of wind turbine performance (e.g. Wagner et al., 2009), understanding wake interactions with the ABL in large wind farms (Kumer et al., 2015;





Lungo, 2016; Li et al., 2016) and as boundary conditions for simulations of gas dispersion in the ABL (e.g. Labovský and
Jelemenský, 2011). Popular systems for getting the required wind velocity data in different regions of the ABL are traditional
mast-mounted anemometers (mostly cup and sonic anemometers, e.g. Izumi and Barad, 1970), balloons (e.g. Scoggins, 1965),
sonic detecting and ranging (SODAR, Reitebuch and Emeis, 1998) and light detection and ranging (lidar, e.g. Bilbro et al.,
1984). The methods are suitable for measurements of different ranges of temporal and spatial scales, they result in comple-
mentary data and are often subject to comparisons (e.g. Barthelmie et al., 2014).

## 1.2   Unmanned aerial vehicles as sensor platforms

Despite the large variation of existing measurement techniques, there still is a gap in the wind data collection in the ABL,
driving the development of small unmanned aerial vehicles (UAVs) that are equipped with sensors measuring wind velocities
(Ivey et al., 2014; Elston et al., 2014; Lauer and Fengler, 2017; Prudden et al., 2018; Rautenberg et al., 2018; Barbieri et al.,
2019). These UAVs are less suitable for long-term measurements as the endurance is typically limited. However, they have
three-dimensional (3D) mobility, are easy to deploy, inexpensive and flexible to operate, and provide time and space-resolved
measurements of wind speeds (e.g. Nichols et al., 2017). They are versatile and can easily be equipped with additional sensors
to map further air properties in 3D space. UAVs are therefore believed to contribute valuable data to research in the ABL (e.g.
Van den Kroonenberg et al., 2008; Wildmann et al., 2014; Prudden et al., 2018; Rautenberg et al., 2018; Barbieri et al., 2019).

There are two fundamental types that are suitable for atmospheric wind measurements: fixed, and rotary wing UAVs. In fixed
wing UAVs, the wings provide the force ($L$) to counter weight ($W$). Propellers supply thrust ($T$) that is needed to overcome the
drag ($D$) of the aircraft, hence thrust is proportional to $W/(L/D)$. In contrast, rotary wing UAVs provide all the force that is
needed to offset weight by the thrust of propellers, hence $T$ is proportional to $W$. As the lift-to-drag ratio ($L/D$) of fixed wing
UAVs can easily exceed unity and often reaches values of about 10 (Thielicke, 2014), the thrust requirement of similar-sized
fixed-wing UAVs is significantly smaller. Power is thrust times speed, hence the power requirement of fixed-wing UAVs can

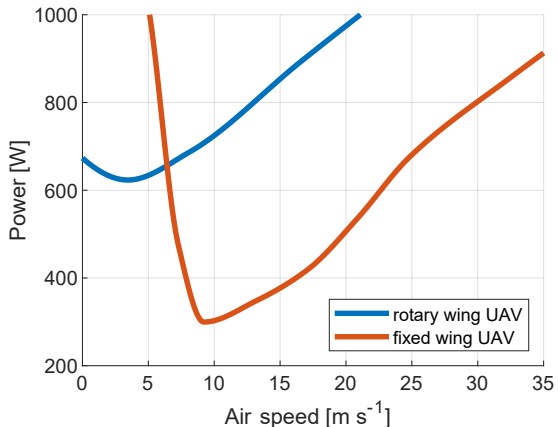

**Figure 1.** Power requirement of fixed vs. rotary wing UAVs with the same weight, (redrawn from Poh and Poh, 2014).





be much lower than in rotary-wing UAVs at similar flight speeds (see Figure 1). This makes fixed-wing UAVs very suitable

for measuring tasks that require long flight times and large areas to be covered (Thielicke, 2014), especially in areas with high wind speeds that require elevated air speeds of the vehicle.

Rotary wing UAVs have a lower endurance, typically by a factor < 0.5. They are however easier to deploy and operate, have the capability to hover on the spot and are much more manoeuvrable (Thielicke, 2014). Therefore, they can measure wind speeds close to structures and perform measurements at a single spot for prolonged periods. These are desirable properties e.g.

when validating wind measurements of the UAV in proximity to traditional anemometers in the field.

### 1.3 UAV-based wind speed measurements

Wind can be determined using two different approaches in UAVs: Indirect methods measure the response of the UAV to the wind and can determine wind speed, azimuth and elevation directly from the sensors that are also used to control the UAV (e.g. Neumann et al., 2010; Johansen et al., 2015; Xiang et al., 2016; Lauer and Fengler, 2017; Donnell et al., 2018). Such

methods require knowledge about the inertia and drag coefficients of the UAV in a larger set of situations. These are not trivial to determine but have a large impact on the accuracy of wind measurements. Using this approach can increase the flight time and the maximum sustainable wind speed, as no additional sensor must be carried.

Direct methods use dedicated wind sensors that are mounted to the UAV. Suitable sensors should be light weight, robust and measure a 3D wind vector. Together with the 3D ground speed vector of the UAV, wind speed and direction can be derived

by simple vector addition. If the sensor is mounted on a gimbal, ensuring zero pitch and roll angle during flight, then a two-dimensional sensor can be sufficient to measure 2D wind speed. In practice, the fusion of vehicle speed and wind speed can yield erroneous results due to errors in wind sensing and vehicle state estimation. These errors become visible as periodic signals in the wind data having a similar frequency as the vehicle speed, attitude, or position (Nichols et al., 2017).

Several types of sensors have already been used to measure wind speeds with fixed and rotary wing UAVs. Using differential

pressure sensors like pitot tubes and multi-hole pressure probes is most suitable when the wind measurement covers larger areas and the UAV is constantly moving forward at elevated speeds. They require the wind to come mainly from one direction within the cone of acceptance of the probe. Furthermore, these sensors perform best at speeds $> 3\,\mathrm{m\,s^{-1}}$ (Prudden et al., 2018), and can provide accurate wind speed measurements with high frequency (typically $> 1\,\mathrm{kHz}$). Differential pressure sensors have been successfully mounted to fixed wind UAVs (for an overview, see Rautenberg et al., 2018) and to rotary wing UAVs (e.g.

Prudden et al., 2016).

Mechanical anemometers (e.g. cup anemometers) are rarely seen on UAVs. Most mechanical anemometers cannot be used for accurate measurements of a 3D wind vector. Arrays of several sensors in different orientations would have to be analysed. This leads to a bulky setup and a large measurement volume. The response time is generally low (Camp et al., 1970), making them less suitable for measurements in rapidly changing environments such as a flying UAV. The most promising mechanical

anemometer with 3D wind sensing capabilities appears to be the k-Gill propeller vane anemometer (e.g. Bottma et al., 1995), it has however not been implemented on UAVs to date.





Very recently, a lidar sensor has been mounted to a rotary wing UAV (Vasiljević et al., 2020). This enables the UAV to precisely measure wind at remote locations and extremely close to structures without the risk of collision. Wind measurements are reported to be in excellent agreement with reference sensors. Currently the lidar is split into an airborne part (telescopes), and

a ground-based part (optoelectronics), interconnected by optical fibers which currently limit the cost effectiveness, deployment radius and maximum altitude.

Hotwire anemometers can be used for high frequency wind speed measurements and have a flat frequency response up to 7 kHz (Hutchins et al., 2015). They have a small measurement volume and can measure 3D wind speeds, making them a good choice for accurate flow measurements. However, the fragility of these sensors increases handling difficulty of a rapid-to-deploy

UAV in harsh environments.

Sonic anemometers that are attached to rotary wing UAVs have been shown to provide accurate wind measurements (Barbieri et al., 2019). In the past, full-sized sonic anemometers were rarely applied to UAVs due to size and weight constraints (Elston et al., 2014). An exception is presented by Donnell et al. (2018) and Natalie and Jacob (2019), where a full-sized 1.7 kg sonic sensor (R.M. Young 81000 3D ultrasonic anemometer) has been attached to a large, 12 kg commercial multirotor (DJI M600).

However, the data from Natalie and Jacob (2019) show that wind speed measurement under challenging conditions is off by up to 50% and azimuth is off by 15° with respect to a lidar reference.

In the study from Donnell et al. (2018), measurements were averaged in bins of 10 s, the resulting wind speed has a bias of 16% and a root-mean-square-error (RMSE) of 46%. The wind azimuth has a bias of -59° with an RMSE of 114°. Another ground-based Young Model 81000 sonic anemometer was used as reference instrument. Although the data sheet of the sonic

anemometer that was used on the UAV states vertical acceptance angles of ± 60°, the accuracy at highly non-horizontal inflow has not been examined in a wind tunnel in this study. The performance at non-zero angles of attack must be validated with great care for sonic anemometers, as error in wind speed measurements can easily reach 20% (Christen et al., 2001; Nakai et al., 2006; Kochendorfer et al., 2012; Nakai and Shimoyama, 2012). Furthermore, sensors that are mounted at a significant distance to the propeller disk avoid interference from induced flows: Wind tunnel studies from Prudden et al. (2016) imply a

distance of about 3 rotor diameters to significantly reduce the effect of induced flows. A distance of 0.97 rotor diameters was used in the two studies mentioned above. These circumstances might explain the relatively large measurement uncertainty in Donnell et al. (2018) and Natalie and Jacob (2019) despite the use of an industry standard sonic anemometer.

Lately, several small-sized sonic anemometer have become commercially available (e.g. Decagon Devices DS-2, Anemoment TriSonica, FT Technologies FT205), which have subsequently been successfully applied to UAVs (Palomaki et al., 2017;

Donnell et al., 2018; Nolan et al., 2018; Hollenbeck et al., 2018; Natalie and Jacob, 2019; Adkins et al., 2020).

Palomaki et al. (2017) used a small 2D sensor (Decagon Devices DS-2) on a DJI Flame Wheel F550 hexrotor. The sensor is mounted 0.8 rotor diameters away from the rotor disks. The bias error resulting from the propeller induced flow was determined for zero-wind condition and subtracted from the anemometer output. According to the manual (Decagon Devices, Inc, 2017), this anemometer needs to be levelled precisely to provide accurate wind speed measurements. This will not be the case on a

multirotor when wind speed is ≠ 0. However, the study reports very low uncertainties in wind speed (2.6% bias, 16% RMSE) and wind azimuth (8.8° bias, 39° RMSE).



In an additional experiment, the study of Donnell et al. (2018) used a Trisonica mini sonic anemometer mounted 0.8 rotor diameters above the propeller disks of a 3DR Solo quadrotor. The study performed wind tunnel measurements with the sonic anemometer, showing an average error of only 3%. However, the sensor was tested at horizontal inflow only and the sensitivity

to changes in wind azimuth angle was not assessed. When mounted to the quadrotor, wind speed measurements differed between 20 and 50 % from the reference measurement.

Nolan et al. (2018) use a DJI Inspire 2 quadrotor with a two-dimensional Atmos 22 sonic anemometer which - according to the datasheet - should remain levelled during measurements (METER Group, 2020). The sensor is mounted at 0.8 rotor diameters, reported wind speeds differ on average about 15% from the reference.

Hollenbeck et al. (2018) used the Trisonica mini on a Foxtech Hover1 quadrotor at 1 rotor diameter from the propeller disk. Wind tunnel measurements were conducted at a single yaw angle and zero pitch angle, showing an error in wind speed of 10 - 20% of the sensor alone. The effect of rotor-induced flows was assessed to be negligible (based on smoke trail visualization with running motors in a wind tunnel). However, these induced flow tests were conducted with zero pitch angle of the quadrotor, which is only representative for zero airspeed. No data on the accuracy of wind speed measurements during free flight is

available.

Adkins et al. (2020) attached two FT205 (FT Technologies Ltd, 2020) miniature sensors at right angle to a large drone (DJI S1000). They were mounted at about 2 rotor diameters from the propellers. There is no data on the accuracy of wind speed measurements available yet, however, the setup seems promising, as true 3D wind speed information can be derived from these two 2D anemometers.

The number of applications of miniature sonic sensors on UAVs is growing (e.g. Barbieri et al., 2019). Accurate 3D wind measurements seem challenging with miniature sensors, as shadowing effects, that are already problematic in full-sized sonic anemometers (Grare et al., 2016), will play an important role as soon as the UAV is not flying perfectly levelled anymore. We believe that great care must be taken during the calibration and validation of miniature sonic anemometers.



## 2 System design

### 2.1 General approach

Due to the simplicity of deployment, the ability to measure close to structures and the potential to uninterruptedly fly the UAV via power-tethering, we decided to use a rotary wing UAV as platform. Commercial, of the shelf (COTS) wind measuring drones are not yet available. Several studies, including the ones mentioned above, use COTS drones (e.g. by companies such as DJI, 3DR, Yuneec) to carry the sensor payload. However, the flight time of a drone can only be optimized for a specific payload

weight. Most COTS drones with sufficient endurance (> 30 min) are designed for larger payloads (> 1 kg) and have take-off weights easily exceeding 10 kg. We therefore designed a custom quadrotor drone around a well-proven, highly customizable, open source flight controller (ardupilot.org), enabling us to combine a custom frame with appropriate COTS electronic components and a suitable wind sensor. Keeping the total weight below 5 kg - which reduces the amount of required administrative decisions for take-off, and a long flight time (> 45 minutes) were on top of the list of requirements.

We identified sonic anemometers to be most suitable for the application in rotary-wing UAVs. These anemometers can sense wind from any azimuth angle from zero speed to about $50 \, \mathrm{m \, s^{-1}}$. The vertical acceptance angle is up to $30°$ for some models. Rotary wing UAVs are manoeuvrable because they can move and rotate almost without restrictions in 3D space. Therefore, omnidirectional wind measurements are important to keep this benefit in manoeuvrability. Several sensors are available as COTS components, some come pre-calibrated to compensate for inbuilt shadowing effects.

Based on the literature review presented above, we believe that special attention must be paid for the following parameters, when designing an accurate drone-based wind measuring system:

1. Accuracy in 3D flow of the sonic anemometer (mini and full-size)

2. Maximization of endurance via weight minimization

3. Accuracy of the data fusion with the UAVs attitude and speed

4. Sensor placement: influence of propeller-induced air flow

5. Accuracy of the full measurement system

6. Practicability of the measurement system in the field

The following sections describe how we analysed these parameters for our flying anemometer. We studied the 3D sensing performance of a miniature sonic anemometer and a pre-calibrated full-size sensor (with removed post to reduce weight and

moment of inertia) in a calibration wind tunnel. Additionally, we analysed the influence of the propeller-induced flow by flying the UAV with attached anemometer inside a large wind tunnel. We also validated that there is no crosstalk between ground speed and wind speed during flight. The accuracy of the drone-based measurements was validated at several altitudes with a bistatic lidar. Finally, we tested the UAV in a typical measurement campaign: Wind turbine wakes are usually mapped using





lidar (e.g. Smalikho et al., 2013; Wu et al., 2016; Herges et al., 2017), which is relatively cost intensive and laborious. We

tested the feasibility of UAV based measurements in the field by flying in the wake of a wind turbine in complex terrain.

## 2.2   Drone design

The drone was composed from COTS electronic components and a custom frame. The aim was to achieve the maximum flight time for the specified sensor payloads while keeping a total weight below 5 kg. The design was based on calculations using available motor and propeller performance data from different manufacturers. After several iterations, suitable propellers and

motors were found, frame weight and payload weight could be determined and refined using preliminary CAD drawings. A suitable battery that maximizes flight time but keeps the weight below 5 kg was finally selected. The frame consists of wound carbon fibre tubes and sandwich carbon sheets with balsa end grain core, and some 3D printed covers. See the data sheet (see Section 'Data availability') for detailed information on all the components that were used.

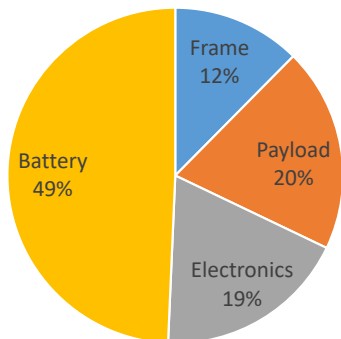

**Figure 2.** Weight break down of the drone components.

Air speeds of up to $20 \, \mathrm{m \, s^{-1}}$ have been successfully tested in flight. The relation between air speed and pitch angle is shown

in Figure 4. A pitch angle of $30°$ is not exceeded in normal forward flight. The power consumption has a minimum at $7 \, \mathrm{m \, s^{-1}}$. With a 444 Wh battery, we can achieve a theoretical maximum flight time of 54 minutes. In practice, only 85% of the stored energy should be used for safety and battery lifetime reasons, which results in 46 minutes flight time.

## 2.3   Anemometer data fusion with UAV speed, attitude and (angular) velocity

Wind speed can be derived from the sum of the relative wind vector (as measured by the sonic anemometer) and the ground

speed vector (as measured by the UAV). The UAV uses Ardupilot's extended Kalman filter estimation system (EKF2, Pittelkau, 2003, ardupilot.org) which estimates vehicle position, velocity and angular orientation based on gyroscopes, accelerometer, magnetometer, GNSS, barometer and ground distance measurements.

Wind speed is transformed from a body-fixed reference system (BFRS) to the terrestrial reference system (TRS) using standard rotation matrices. We also compensate for the airflow induced by angular velocities in roll and pitch of the UAV. The

input and output data for this transformation is given in Figure 5. All these calculations are performed on an onboard computer





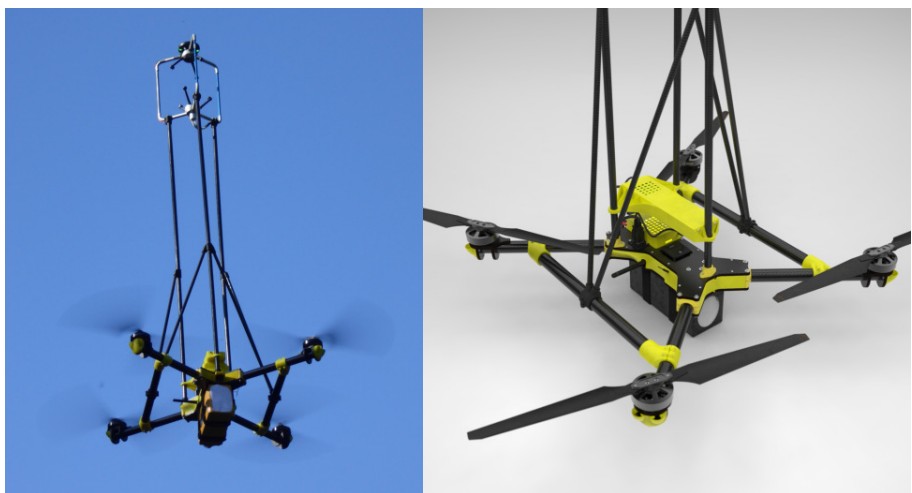

**Figure 3. Left:** The OPTOkopter drone with a Gill Windmaster on a 1 m long mount during a measurement flight. **Right:** Rendering of the CAD model.

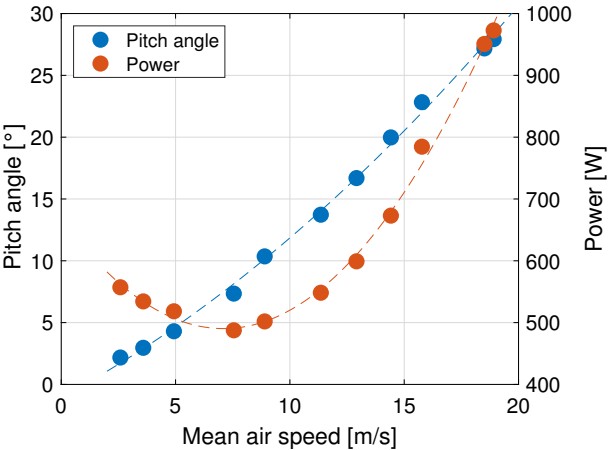

**Figure 4.** Flight properties on the OPTOkopter for flight speeds between 2 and 18 $\mathrm{m\,s^{-1}}$.

(Raspberry Pi 3B+) that is getting wind speed data from the sonic anemometer at 16 Hz and attitude, position, and ground speed information from the flight controller of the UAV at 10 Hz. The onboard computer stores the measurement location in north, east, down terrestrial reference system. The wind speed vector is stored in West-East, South-North, Up-Down terrestrial reference system.





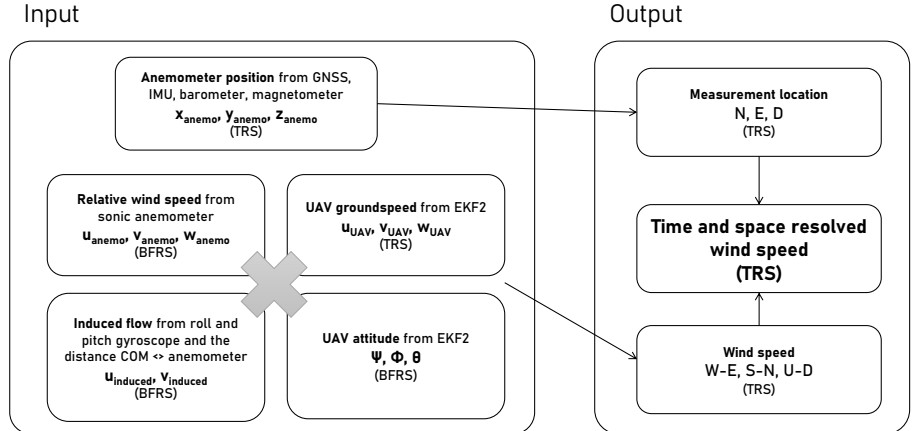

**Figure 5.** Data fusion of sonic anemometry and UAV attitude, position and ground speed. $x_{anemo}$, $y_{anemo}$, $z_{anemo}$ is the position of the anemometer in terrestrial reference system (TRS). $u_{anemo}$, $v_{anemo}$, $w_{anemo}$ is the relative wind speed as measured by the anemometer in body-fixed reference system (BFRS). $u_{UAV}$, $v_{UAV}$, $w_{UAV}$ is the ground speed of the UAV as reported by the EKF2 in TRS. $u_{induced}$ and $v_{induced}$ is the induced flow velocity at the measurement volume of the anemometer (lever arm · angular velocity) for pitch and roll. $\psi, \phi, \theta$ is the attitude of the UAV as reported by the EKF2 in BFRS. The fusion of the data yields the measurement location in North, East, Down (N, E, D; TRS) and the wind speed in West-East, South-North, Up-Down (W-E, S-N, U-D; TRS).

## 3  Validation

### 3.1  3D wind sensing performance of sonic anemometers

A miniature sonic anemometer (TriSonica Mini Wind and Weather Sensor, Anemoment, 2020) and a full-size, sonic anemometer (factory precalibrated Windmaster, Gill Instruments Limited, 2020) were tested at 0 - 360° yaw with 0°, 10°, 20°, 30° pitch angles (see Figures 6 and 7) in a traceable wind tunnel (400 · 260 mm cross-section, 400 mm length; accredited according to ISO/IEC 17025, Eidgenössisches Institut für Metrologie METAS, Switzerland) at wind speeds between $1\,\mathrm{m\,s^{-1}} < \mathrm{v} < 15\,\mathrm{m\,s^{-1}}$. Our measurements were compared with a calibrated propeller anemometer (measurement uncertainty 2%) at 20°C, 950 hPa, 47% humidity. Both anemometers were mounted in the wind tunnel using the same attachments as in the drone, including global navigation satellite system (GNSS) / magnetometer and cable connections to assure that measurement conditions reflect the real situation on our flying drone.

A suitable anemometer for application on UAVs should be able to accurately sense wind speed, azimuth and elevation. This should be possible for all pitch, roll and yaw angles that occur during a typical measurement flight of the UAV. In our design, the maximum pitch angle of the UAV at the maximum air speed ($20\,\mathrm{m\,s^{-1}}$) is 30°. The wind tunnel measurements of the GILL Windmaster anemometer show that bias and RMSE are small, but wind speed is overestimated by up to 3.6% at 30° pitch. Note that the results in Table 1 show bias and RMSE for measurements at $15\,\mathrm{m\,s^{-1}}$ and 0 - 360° yaw angle at four different pitch angles. Wind speed measurements with the Trisonica are lower than the reference. At large pitch angles there is a strong





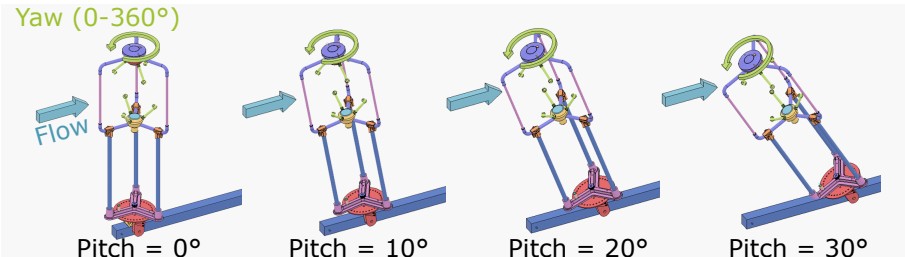

**Figure 6.** Wind tunnel tests of the 3D wind sensing capabilities. Rotation around the yaw axis (green arrow) for four different pitch angles (from left to right: 0°, 10°, 20°, 30°). Wind is coming from the left (blue arrow). A global navigation satellite system (GNSS) receiver is mounted on top of the anemometer.

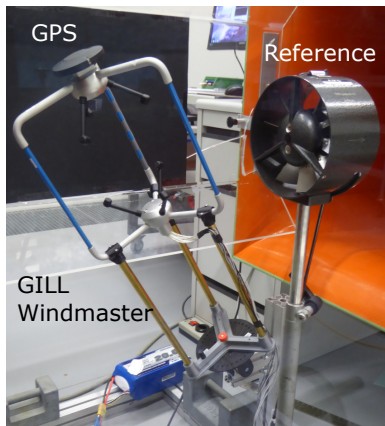

**Figure 7.** Setup of the experiment in the calibration wind tunnel of METAS. Flow is from left to right.

bias (-17.3% with a RMSE of 16.2%). At zero pitch, there still is a bias of -6.3% and a RMSE of 5.4%. Wind azimuth sensing of the Windmaster is almost by a factor of 10 more accurate than in the Trisonica for both bias and RMSE. The Windmaster has a particularly good performance in sensing the wind elevation with a maximum bias of 1.3° and 1.4° RMSE. This is not the case for the Trisonica, where no relation between pitch angle and elevation could be determined.

The Trisonica was tested in the wind tunnel in November 2018. After we reported our results to the manufacturer, a firmware update (ver 1.7.0, February 2019) addressed the issue of wind shadowing, potentially enhancing the accuracy at zero pitch angle. We had no opportunity to test this firmware in a wind tunnel yet. The issue at higher pitch angles will most likely remain, as we think it is impossible to accurately measure a vertical wind component with a small device with an inherently high blockage ratio. The latest firmware still needs to be tested in a wind tunnel for 0-360° and several pitch angles to check

for improvements in accuracy. Based on our measurements, we think that the accuracy of 3D wind measurements with hard-mounted miniature sonic anemometers on UAVs is limited, and might explain the limited accuracy that several studies report for in-flight measurements with these kind of sensors (see Introduction). Miniature sensors should be mounted on a stabilizing





| | Windmaster | | Trisonica | |
| --- | --- | --- | --- | --- |
| | Speed [%] | | Speed [%] | |
| Pitch | Bias | RMSE | Bias | RMSE |
| 0° | 0.1 | 1.4 | -6.3 | 5.4 |
| 10° | 1.5 | 1.7 | -4.5 | 5.1 |
| 20° | 1.8 | 3.1 | -10.6 | 10.2 |
| 30° | 3.6 | 2.7 | -17.3 | 16.2 |
| | Azimuth [°] | | Azimuth [°] | |
| Pitch | Bias | RMSE | Bias | RMSE |
| 0° | -0.6 | 0.7 | -5.9 | 4.4 |
| 10° | -0.4 | 1.0 | -6.2 | 3.4 |
| 20° | -0.2 | 1.7 | -6.3 | 8.5 |
| 30° | -2.8 | 2.7 | -5.1 | 16.5 |
| | Elevation [°] | | Elevation [°] | |
| Pitch | Bias | RMSE | Bias | RMSE |
| 0° | 0.6 | 0.5 | 2.1 | 4.2 |
| 10° | 0.9 | 0.5 | -10.7 | 4.6 |
| 20° | 1.3 | 0.8 | -24.2 | 5.9 |
| 30° | 1.0 | 1.4 | -37.7 | 10.5 |

**Table 1.** Wind tunnel test. Accuracy of a miniature sonic anemometer (Trisonica) and a full-size sonic anemometer (Windmaster). Each pitch angle was tested with 0-360° yaw rotation at 15 m s$^{-1}$ wind speed. Bias and RMSE are based on the measurements of a full yaw rotation.

gimbal and with the UAV flying at constant altitude to ensure that the vertical wind component is kept low. The accuracy of the full-size sonic anemometer is well within the specs given by the manufacturer. Although it weighs by a factor of 20 more
than the miniature sensor, we believe that the full-size anemometer is a more suitable instrument for the highly 3D flow on a flying and manoeuvring non-stationary UAV.





## 3.2 Influence of the propeller induced flow

An anemometer that is mounted on a rotary-wing UAV is potentially measuring a velocity component that is induced by the propellers. It therefore potentially measures a biased wind speed and a biased elevation. The induced component most likely
depends on the forward flight speed (air speed in this case). In normal free flight, every flight speed requires a certain pitch angle and propeller speed. We therefore determined suitable pitch angles and throttle values (voltage sag compensated) for front and rear motors by flying circles (D = 300 m) at different speeds while sampling pitch angle and motor throttle at 10 Hz (see Figure 8). We used this data for measurements in the wind tunnel of the Technische Universität Dresden (open test section, diameter = 3 m). Wind speeds between 1 and 19 $\mathrm{m\,s^{-1}}$ were tested with the OPTOkopter being tethered to a variable pitch mount. The drone was also flown freely inside the wind tunnel to validate that the mount did not influence the measurements.

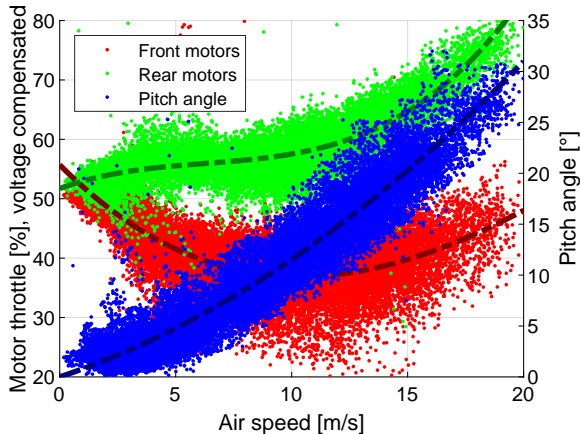

**Figure 8.** Motor throttle (front and rear motor pairs) and pitch angle vs. flight speed. The spread can be explained by the control loops working against external disturbances during free flight in windy conditions. Rear motors generally need to provide more thrust than the front motors to compensate for a pitch-up moment. This effect results from a combination of uneven lift distribution of the rotors in forward flight and gyroscopic precession (US Department of Transportation, 2019). The additional drag of the sonic anemometer amplifies the pitch-up moment which is compensated by higher throttle on the rear motors.


We found only little effect of the propeller flow on the measured wind speed (see Figure 10). The wind speed bias is smaller than 1.5% for wind speeds above 5 $\mathrm{m\,s^{-1}}$. The story is different for the wind elevation, where the propeller induced flow has a large effect for wind speeds $\leq 10\ \mathrm{m\,s^{-1}}$ (see Figure 11). The bias at very low wind speeds reaches $11°$. This is remarkable, because in comparison to previous studies (see Introduction), the OPTOkopter has a large distance between the anemometer
and the propeller disks (1.15 m = 2.5 rotor diameters). The effect diminishes with increasing air speed, however. We compensate for this propeller induced flow using $v_{vertical,corrected} = v_{vertical} - t \cdot 0.007$, where $v_{vertical}$ = vertical velocity component (in TRS); $t$ = average motor throttle in percent. The method keeps the bias of wind elevation below $1°$. The effect of our compensation method is also shown in Figure 11. To conclude, the propellers impact the direction of the flow (air is deviated





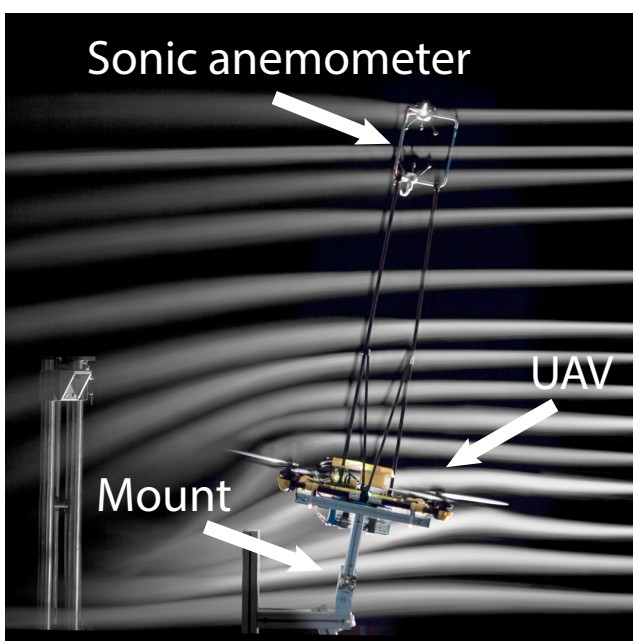

**Figure 9.** Setup of the wind tunnel measurement. The UAV is attached to a mount at multiple suitable pitch angles (here: 10°) representing forward flight. Wind speed measurements (here: $9\,\mathrm{m\,s^{-1}}$) with running motors were compared to measurements with stopped motors. This 'tethered' setup was also compared to free flights in the wind tunnel. Smoke trails were used for illustration purposes. A video of the wind tunnel testing is available at youtu.be/wWPY3eVxUkU.

downwards, which can be effectively compensated), but there is only a small influence (about 1%) on the horizontal speed of
the air, even at high pitch angles.



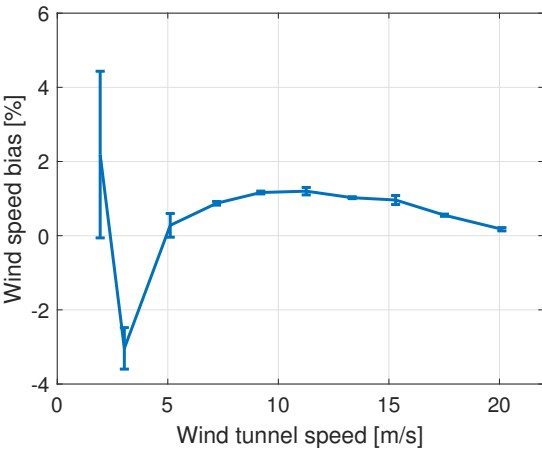

**Figure 10.** Bias of wind speed measurements with running motors, tested in a large wind tunnel. Wind speed measurements with stopped motors are used as reference. The bias is $\leq 1\%$ for wind speeds above $5\,\mathrm{m\,s^{-1}}$.

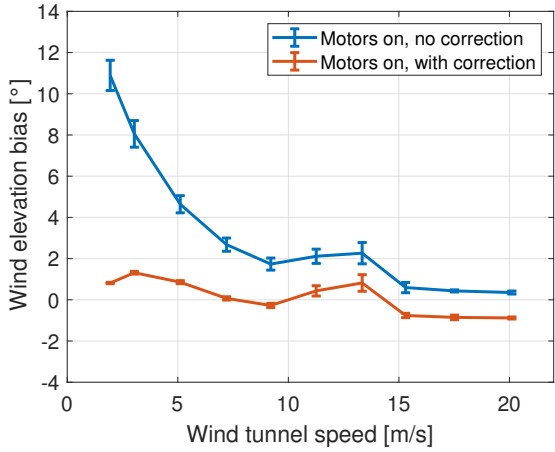

**Figure 11.** Bias of elevation measurements with running motors. Measurements with stopped motors are used as reference. The simple correction algorithm limits the bias to $\leq 1°$.

### 3.3 Crosstalk between ground speed, air speed and wind speed

After testing the performance of individual components in our measurement system, we assessed the accuracy of the full flying setup. As mentioned in the introduction, Nichols et al. (2017) report that periodic signals in the wind estimation can often be seen when a UAV is flying periodic manoeuvres and data fusion is imperfect. We checked for such problems by rapidly flying

the UAV between two points that were 10 meters apart with a sinusoidal ground speed peaking at about $4\,\mathrm{m\,s^{-1}}$. Ground speed (as reported by the flight controller), air speed (as reported by the anemometer) and wind speed (as reported by our data fusion)





was recorded and converted to the frequency domain using fast Fourier transform. Comparing the power spectral densities allows for evaluating the crosstalk between ground speed measurements and wind speed measurements.

In a situation with zero wind, air speed and ground speed as measured by the UAV must be identical. When there is wind,
these velocities will not be identical anymore. But any change in ground speed will also result in a change in air speed, hence, a spectral analysis should show peaks at the same frequencies. This is the case in our test flight (see Figure 12): Both air and ground speed have a peak at 0.208 Hz. This is the frequency that the OPTOkopter was oscillating between two waypoints. A linear regression for ground speed and air speed yields a Pearson's correlation coefficient of 0.78.

The FFT analysis (see Figure 12) reveals, that the power spectral density of the wind speed at the relevant frequency
(0.208 Hz) is 4 to 5 orders of magnitude smaller than air speed or wind speed. Additionally, the correlation coefficient for ground speed and wind speed is 0.004. These analyses indicate that our fusion algorithm results in a wind speed measurement that is independent of ground speed and UAV motion / rotation in general. This is very important for airborne measurement systems that do not only perform point measurements in hovering flight, but are also capable of measuring while flying a mission. Such a measurement is presented in Section 4.

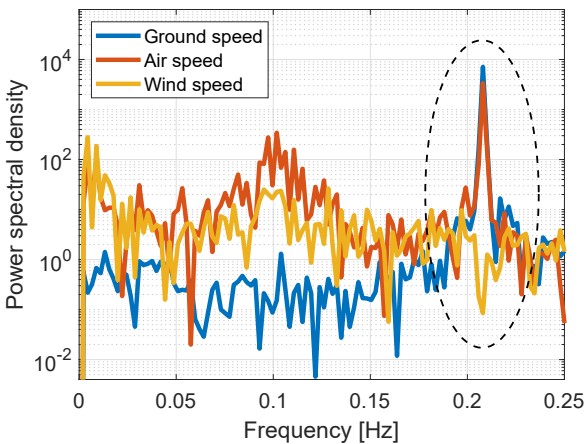

**Figure 12.** Power spectral densities of ground speed, air speed and wind speed during a flight were the UAV was repeatedly oscillating between two waypoints. There is a clear peak at 0.208 Hz in ground speed and air speed, but no peak in wind speed, indicating that wind speed measurements are not affected by the motion of the UAV.



### 3.4 Comparison with a bistatic lidar

We compared our wind measurements with the bistatic Doppler lidar, developed at the Physikalisch-Technische Bundesanstalt (PTB) in Braunschweig, Germany (Oertel et al., 2019; Mauder et al., 2020). A data output rate of 10 Hz was used in the PTB lidar, and different heights between 20 and 100 m were tested.

The bistatic lidar has a small, stationary measurement volume. The distance between the measurement volumes of the OPTOkopter and the lidar was difficult to assess as there was no optical reference that could help with relative positioning as the exact measurement position of the lidar is not visible. However, attempting to fly close to this volume is relatively safe if optical instruments on the UAV such as distance finders and cameras are isolated from the high laser power. After

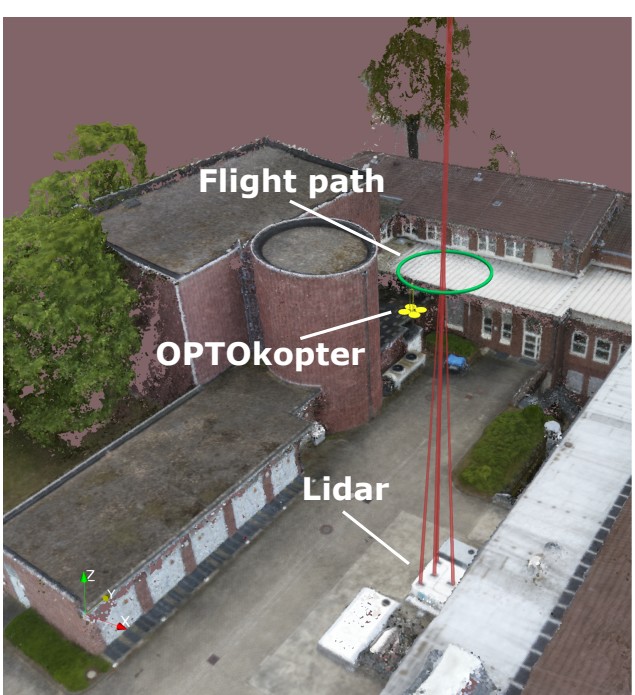

**Figure 13.** Measurement site of the PTB lidar reference. This 3D model of the location was calculated using photogrammetry with image data that were captured during the wind measurement flights.

performing several flights, we selected a measurement at 40 m height for a detailed analysis, as the time-shift between wind speed measurements of the two methods (determined via cross-correlation) was minimal for this dataset, indicating that we were flying very close (within 1 m range) to the measurement volume of the PTB lidar. We compared the data of the lidar reference to our measurement using an orthogonal Deming regression. RMSE and bias (based on paired observations) were determined for all measurement flights.

The OPTOkopter was always hovering at the lee side of the measurement volume (as indicated by a positive time-shift in the cross-correlation signal). Wind speed, azimuth and elevation were sampled during multiple short flights of 10 minutes.



Additionally, we were measuring wind speeds while circling (4 m radius, 2.5 m s⁻¹ flight speed) around the lidar measurement volume to check for non-zero ground-speed related errors (see Figure 13).

The measurement volume of the lidar at 40 m height is surrounded by tall buildings and trees, generating highly unsteady flow: The wind speed varies by 8 m s⁻¹, the azimuth by about 100°, and the elevation by about 67° during this selected measurement flight (see Figures 14, 15 and 16). These numbers emphasize the importance of being capable to measure three-
dimensional wind with a suitable anemometer. Despite the dynamic situation, the comparison with the PTB lidar reference shows an excellent agreement of the three dimensional wind speed (see Figures 14, 15 and 16). Note that these figures show measurements that were taken with 10 Hz sampling rate.

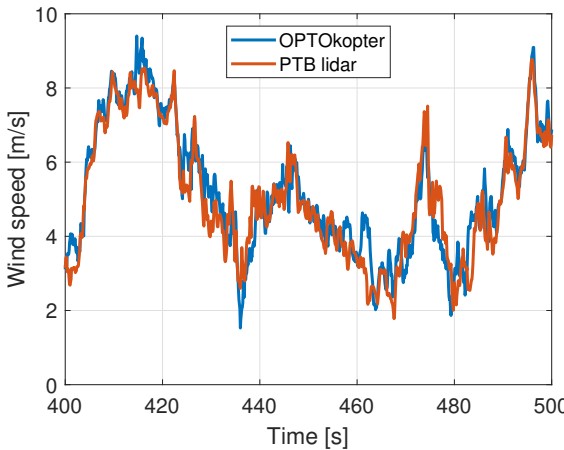

**Figure 14.** Wind speed measurement at 10 Hz, comparison of measurement from the PTB lidar with the OPTOkopter at 40 m height. Wind speed varies between 1.5 and 9.5 m s⁻¹.

A linear Deming regression of the data in 1 s averaging intervals has a slope of 1.03 and an offset of -0.03. The correlation coefficient is 0.95, indicating a good linear relation between both methods (see Figure 17). We also analysed the effect of
increasing averaging intervals (between 0.1 and 100 s) on bias and RMSE. As expected, bias does hardly change when the averaging interval is increased. For the wind speed measurement, bias is between 2.9 and 3.7 %, which is very similar to what we determined for the isolated Windmaster in the calibration wind tunnel (see Table 1). The wind speed RMSE is strongly dependent on averaging interval: It drops from 12% at 0.1 s to 1% at 100 s (see Figure 18).

The azimuth has a constant bias of about 2.6° and a RMSE decreasing from 7° to 1.9° (see Figure 19). The offset can result
from a misalignment of the lidar, or interference of the compass on the UAV. We believe that this uncertainty is acceptable. It could be improved by fusing the heading measurements of the UAV with additional sensors like dual RTK GPS rovers. The elevation bias is constantly at about 0.4°. RMSE decreases from 7° at 0.1 s averaging interval to 1° at 100 s (see Figure 20).

The bistatic PTB lidar and our OPTOkopter hence give closely matched results, even at 10 Hz sampling interval. Naturally, this consistency increases with longer averaging intervals. When measuring at slightly different locations, the influence of spa-

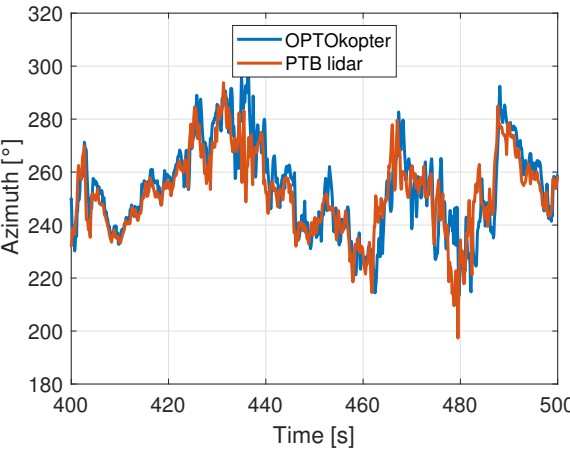

**Figure 15.** Wind azimuth measurement at 10 Hz, comparison of measurement from the PTB lidar with the OPTOkopter at 40 m height. Wind azimuth varies between 200 and 300°.

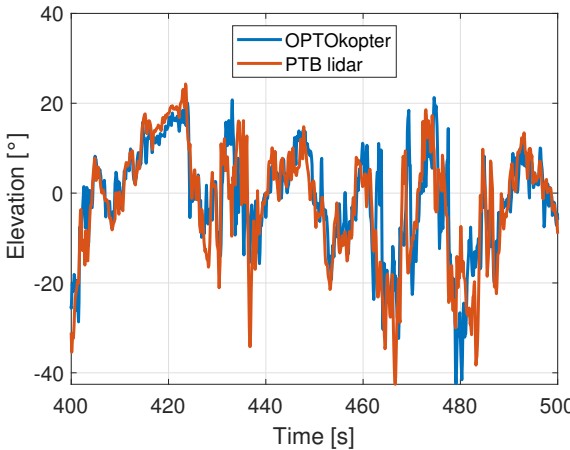

**Figure 16.** Wind elevation measurement at 10 Hz, comparison of measurement from the PTB lidar with the OPTOkopter at 40 m height. Wind elevation varies between -42 and +25°.

tial and temporal wind speed differences decreases with longer averaging intervals, lowering RMSE. Long averaging intervals also reduce measurement noise of both methods, again decreasing RMSE.

Turbulence intensity (averaging interval = 10 s) decreases with altitude (e.g. Svensson et al., 2019). Despite a short sampling time per height (10 minutes) we also find this relation in our measurements (Pearson's r = -0.79, we take the average of $TI_{Lidar}$ and $TI_{UAV}$ to approximate the true TI). Turbulence intensity also shows a significant correlation with wind speed RMSE (r =

0.76), azimuth RMSE (r = 0.90) and elevation RMSE (r = 0.87). As a matter of course, TI also decreases when the averaging interval is increased, yielding lower RMSE (see Figures 18, 19, 20). To conclude:





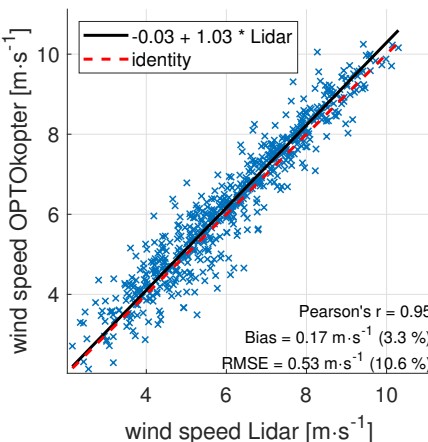

**Figure 17.** Linear Deming regression of the UAV wind speed measurement at 40 m height for 1 s averaging interval. The PTB lidar is used as reference instrument.

- Conditions for high RMSE: ($\Uparrow \Delta t$ **AND** $\Uparrow TI$) **OR** $\Downarrow m_{precision}$

- Conditions for low RMSE: ($\Downarrow \Delta t$ **OR** $\Downarrow TI$) **AND** $\Uparrow m_{precision}$

with $\Delta t$ = time lag between measurement volumes = distance divided by mean wind speed, TI = turbulence intensity, $m_{precision}$ = measurement precision (inverse of random error).

Despite sometimes we were flying very close to the lidar, we believe that the presence of the UAV did not significantly change the flow in the measurement volume of the lidar: The measurements of the OPTOkopter have been successfully compensated for propeller induced flow (see Figure 10 and 11). If the OPTOkopter would have changed e.g. the vertical flow component in the measurement volume of the lidar, then there would be a large discrepancy between (compensated) OPTOkopter measurement and (uncompensated) lidar measurement. Furthermore, measurements at more remote locations that cannot influence the measurement of the lidar due to spatial separation (see Table 2 for bias and RMSE of all tests we performed) have a very similar bias and RMSE.





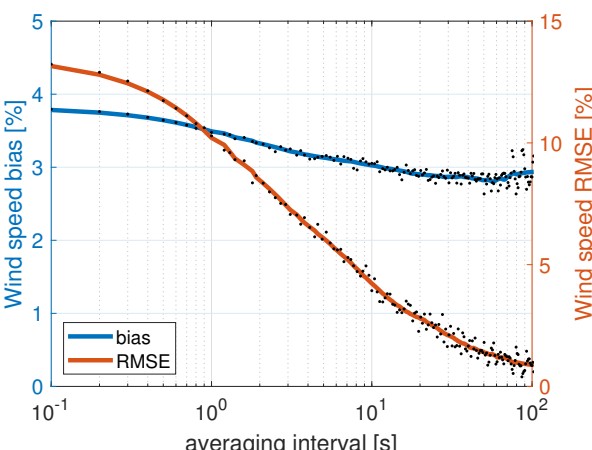

**Figure 18.** Bias and RMSE of the UAV wind speed measurement at 40 m height for averaging intervals between 0.1 s (10 Hz) and 100 s (0.01 Hz). The PTB lidar is used as reference instrument.

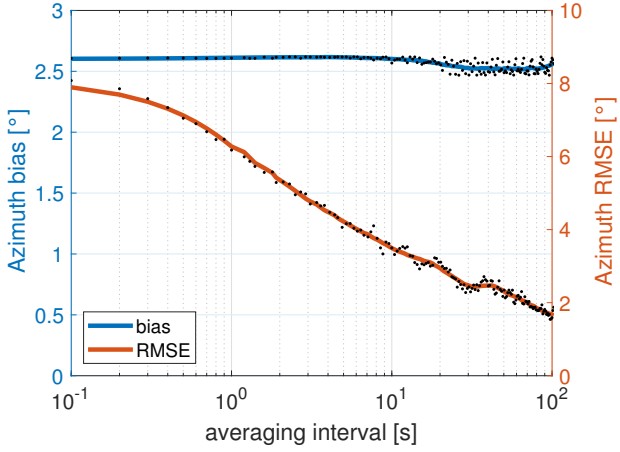

**Figure 19.** Bias and RMSE of the UAV wind azimuth measurement at 40 m height for averaging intervals between 0.1 s (10 Hz) and 100 s (0.01 Hz). The PTB lidar is used as reference instrument.

## 4 Example application: Measurement of a wind turbine wake in complex terrain

The wind turbine (Enercon E 70 - E 4) is located in the Black Forest in southern Germany (47°45'53.43"N; 7°51'11.68"E) at

about 1012 m above sea level. The nacelle height is 85 m and the rotor diameter (D) is 71 m (see Figure 21). Wind velocity was determined at 2 D behind the rotor disk. Flight duration was 22 minutes, and measurements were taken at 16 Hz. The OPTOkopter was oscillating at constant altitude at nacelle height with a velocity of 5 m s$^{-1}$ on a path parallel to the rotor disk (see Figure 21 and 23). Because the wind speed was quite substantially varying with time (see Figure 22), all wind





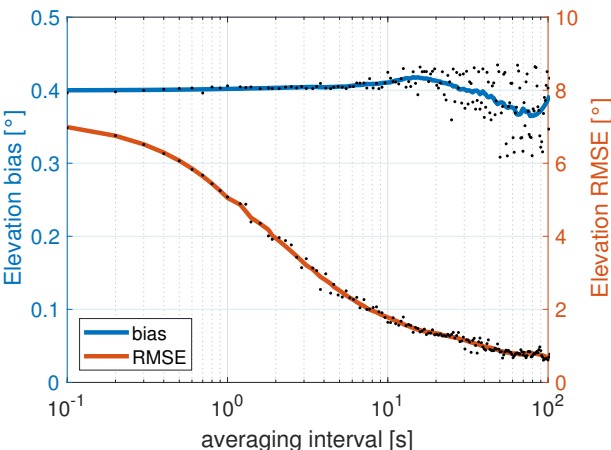

**Figure 20.** Bias and RMSE of the UAV wind elevation measurement at 40 m height for averaging intervals between 0.1 s (10 Hz) and 100 s (0.01 Hz). The PTB lidar is used as reference instrument.

| Altitude | Flight mode | Mean wind speed | | Wind speed | | Azimuth | | Elevation | | Turbulence intensity | |
|---|---|---|---|---|---|---|---|---|---|---|---|
| | | Lidar [m/s] | UAV [m/s] | Bias [%] | RMSE [%] | Bias [°] | RMSE [°] | Bias [°] | RMSE [°] | Lidar | UAV |
| 20 | Circling | 3.7 | 3.7 | 1.3 | 12.4 | -2.6 | 6.7 | -0.6 | 4.7 | 0.29 | 0.28 |
| 20 | Hovering | 3.1 | 3.3 | 3.9 | 10.2 | 7.2 | 8.0 | -1.0 | 5.0 | 0.35 | 0.36 |
| 30 | Circling | 4.4 | 4.4 | 0.9 | 12.7 | 0.3 | 7.6 | 0.7 | 6.3 | 0.33 | 0.30 |
| 30 | Hovering | 4.1 | 4.2 | 1.2 | 11.6 | 6.3 | 6.7 | 0.7 | 5.3 | 0.28 | 0.29 |
| 40 | Circling | 5.1 | 5.2 | 4.2 | 15.5 | -1.5 | 6.2 | -0.4 | 4.4 | 0.31 | 0.29 |
| 40 | Hovering | 6.0 | 6.2 | 2.8 | 4.6 | 3.0 | 3.3 | 0.3 | 1.9 | 0.24 | 0.25 |
| 60 | Circling | 6.7 | 6.6 | -2.0 | 6.4 | -1.9 | 4.3 | -0.1 | 2.4 | 0.18 | 0.18 |
| 60 | Hovering | 8.0 | 7.9 | -0.4 | 4.3 | 3.9 | 2.6 | -0.1 | 1.2 | 0.13 | 0.14 |
| 80 | Circling | 6.5 | 6.6 | 1.3 | 7.8 | -1.4 | 3.8 | 0.4 | 2.6 | 0.19 | 0.18 |
| 80 | Hovering | 7.2 | 7.5 | 4.1 | 6.0 | 6.4 | 3.2 | 0.0 | 2.5 | 0.26 | 0.25 |
| 100 | Hovering | 7.3 | 7.2 | -0.4 | 7.1 | -1.6 | 4.8 | -0.3 | 1.9 | 0.18 | 0.20 |

**Table 2.** Bias and RMSE of the OPTOkopter wind measurements at 10 s averaging interval, with the PTB lidar reference. The table includes data from all flights that were done. The distance to the measurement volume of the lidar was difficult to assess, but it was smaller than 10 m in all cases. The comparison was done with the OPTOkopter hovering on spot or circling around the lidar measurement volume. Wind speed bias is generally low. RMSEs seem to increase with turbulence intensity. Average absolute bias of all the measurements: wind speed 2.0%, elevation 0.4°, azimuth 3.3°. Average absolute bias of hovering flights: wind speed 2.1%, elevation 0.4°, azimuth 4.7°. Average absolute bias of circling flights: wind speed 1.9%, elevation 0.5°, azimuth 1.5°.

measurements were normalized with the reference anemometer velocity on top of the nacelle ($u_{ref}$). Measurements were
discretized in intervals of 1 m along the flight path. Data from each of these bins was averaged.

A relatively constant velocity deficit ($\overline{u}/u_{ref}$) of 25% is found behind the full diameter of the rotor disk. Further away from the rotor tips, the velocity becomes even larger than $u_{ref}$ (see Figure 24). Most likely, the reference anemometer is measuring velocities lower than the true free stream velocity, due to the proximity to the nacelle, and possible shadowing effects by the rotor blades. When a wind turbine rotates clockwise (as viewed from the front), it will generate a swirl with anti-clockwise

rotation. In a horizontal cross-section at nacelle height, this will result in air travelling down on the left side (again viewed from




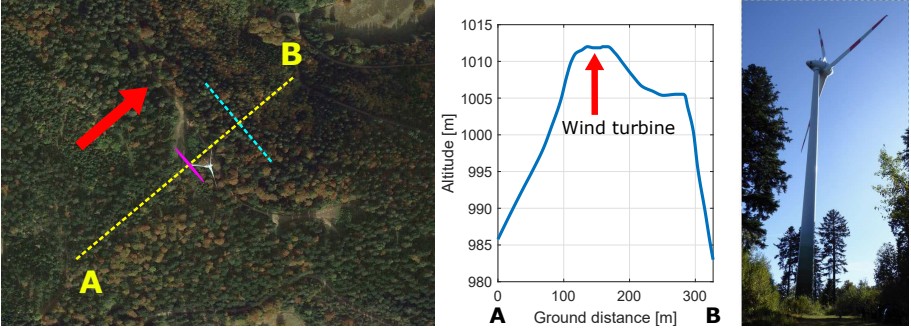

**Figure 21.** Measurement site of the wind turbine study. **Left:** Red arrow: Wind direction. Pink line: Rotor disk in top view. Dashed cyan line: Constant altitude flight path of the OPTOkopter. Dashed yellow line: Path for the elevation profile shown in the middle (image taken from Google Earth) **Middle:** Elevation profile of the measurement site. **Right:** Photograph of the wind turbine in the Black Forest.

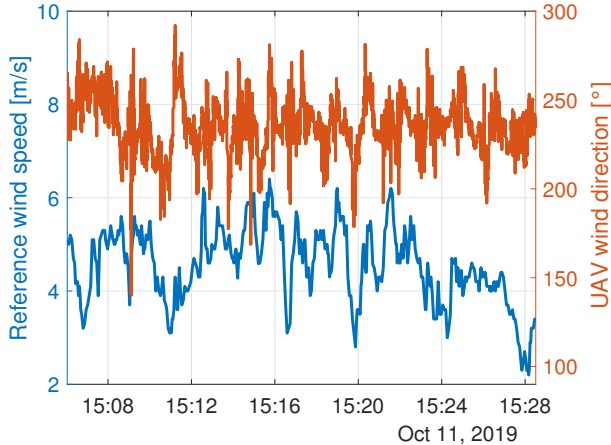

**Figure 22.** Wind speed and wind direction during the measurement flight. Wind speed was measured by the reference cup anemometer on top of the nacelle $u_{ref}$. Wind speed varied between 2.1 and $6.5\,\mathrm{m\,s^{-1}}$. No reference sensor was available for wind direction, hence the figure shows wind direction as measured by the UAV (including all variations introduced by the wind turbine rotor). Wind direction varied between 140 and 290°.

the front), and air travelling up on the right side. We captured the swirl (see Figure 24, right), the magnitude is about $0.35\,\mathrm{m\,s^{-1}}$ which is about 7.7% of the average free stream velocity. The downwash is not perfectly symmetric around the centre of the wind turbine and may be influenced by the slope behind the wind turbine (see Figure 21, middle).

We did not capture data at $> 1$ z/D, as the wind direction slightly changed after the waypoints were positioned and uploaded to the UAV. Our measurements are strikingly similar to theoretical velocity distributions (e.g. Wu and Porté-Agel, 2012; Keane et al., 2016) and lidar measurements in the wake of wind turbines (e.g. Vollmer et al., 2017; Menke et al., 2018). We believe



that the noise in the measurements is mostly due to the inconsistent free stream velocity (see Figure 22) and that it can be decreased significantly by measuring for a longer duration (e.g. using more than one battery pack).

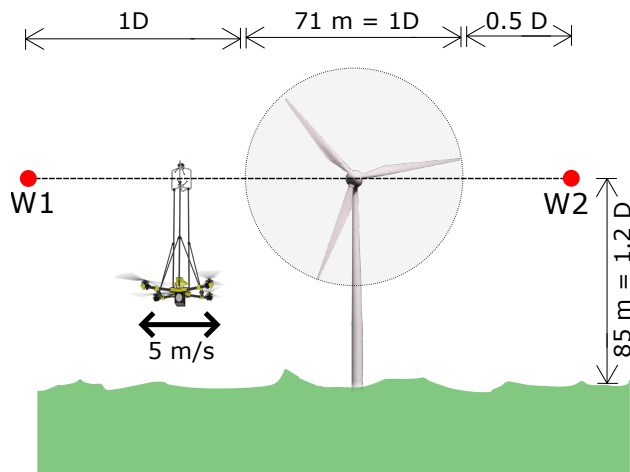

**Figure 23.** Flight path of the OPTOkopter behind the wind turbine (view from the front). The drone oscillates with $5\ \mathrm{m\,s^{-1}}$ between waypoints W1 and W2.

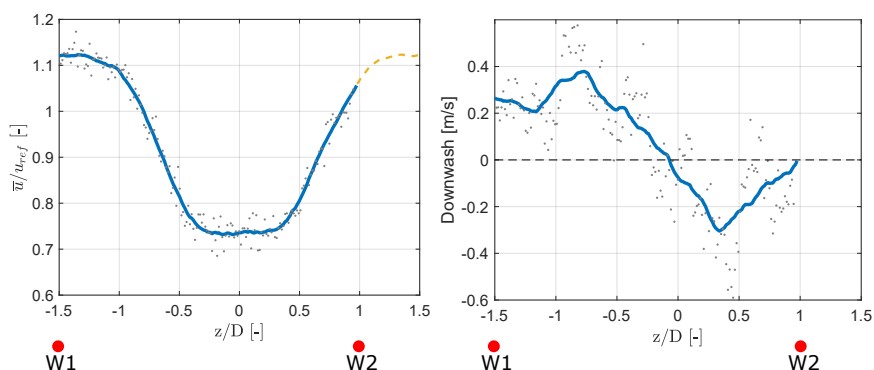

**Figure 24.** Horizontal cross-section at nacelle height through the wake of a wind turbine. **Left:** Points: Discretized wind velocity measurements. Line: Smoothed data (Savitzky–Golay filter). Dashed yellow line: Missing data. A velocity deficit behind the rotor is clearly visible. **Right:** Points: Discretized downwash velocity measurements. Line: Smoothed data (Moving mean). Swirl results in downwash on the left side and upwash on the right side in the wake of the wind turbine.





## 5 Conclusions

The environmental science of the atmospheric boundary layer benefits from wind speed measurements collected by UAVs. We designed a suitable light weight rotary wing UAV for carrying an anemometer. Drones can measure close to structures and they can be validated comfortably by hovering close to a reference instrument. Flight time is often an issue with UAV based measurements. In our design, the battery is responsible for 49% of the total weight. It can be replaced by COTS power-tethering devices, that allow for much longer, uninterrupted measurement flights at a single location at different altitudes up to 100 m.

The OPTOkopter uses a full-size, industry standard anemometer instead of a miniature version, as the accuracy in three-dimensional flow is by a magnitude better. Measurements at the test site of the PTB lidar have shown that three-dimensional flow is highly likely to happen in situ, even when the OPTOkopter hovers on spot at a constant altitude. Due to the high contribution of vertical flow, using a single miniature sonic anemometer does not seem to be feasible on a drone, even when the sensor is mounted on a stabilizing gimbal.

Propeller-induced flow mainly adds a vertical component to the flow without adding a horizontal component - even at large pitch angles. The vertical component can effectively be compensated by subtracting a value that is proportional to the mean motor throttle. We could not find any crosstalk between ground speed and wind speed, although we were flying relatively aggressive manoeuvres (oscillating between two waypoints that were only 10 m apart). These results are supported by exceptionally low bias and RMS errors during the comparison with the bistatic PTB lidar in hovering and circling flight mode (wind

speed average absolute bias = 2.0%, elevation average absolute bias = 0.4°, and azimuth average absolute bias = 3.3°).

Our analysis of the wind velocity in the wake of a wind turbine has proven the practicability of accurate UAV based measurements. The application is not limited to point measurements. The mean wind speed on a 200 m long path behind the wind turbine rotor has been sampled with 1 m resolution. Such an analysis can be executed in significantly less than an hour time including all preparations. The only requirements are a free space of 2·2 meters for take-off and landing (e.g. the roof of a car),

peak wind speeds that do not exceed $20 \, \mathrm{m \, s^{-1}}$, free line of sight between pilot and UAV, and preferably no rain.

Based on our tests of the individual components and the full system, we think that the total measurement accuracy is equal to the accuracy of the anemometer alone, as tested in the wind tunnel. Wind speed and elevation are sensed accurately, when data fusion is performed as described, and separation between wind sensor and propellers is large enough (here: 2.5 rotor diameters). Additionally, the maximum tilt of the drone must not exceed the maximum acceptance angle of the anemometer

(30° in our case).

There certainly is room for improvement in sensing the azimuth (average absolute bias = 3.3°, maximum bias = 7.2°), which is currently limited by the magnetometer. The strongest source of interference usually are the motors of the UAV. We effectively limited this internal interference by mounting the magnetometer at a distance of 1.3 m to the motors. However, external disturbances can occur when flying close to metallic structures, which may still bias the azimuth. More accurate

magnetometers recently became available for use in the Ardupilot firmware (e.g. PNI RM3100). Dual RTK GPSs or landmarks on the ground that are captured from the drone are additional possibilities to reduce azimuth bias.



We think that devices that are designed following the propositions presented in this study, are very suitable for accurate wind measurements up to $20\ \mathrm{m\,s^{-1}}$ and up to 46 min duration. Additional sensors can be attached, e.g. allowing to trace the sources of pollutions automatically, by always flying against the wind vector. Drones like the OPTOkopter, that are specifically designed for the application, might be used as cost effective, flexible and quickly deployable addition to measurement masts and lidar scans.

*Data availability.* Data of all measurements presented in this paper and additional information on the OPTOkopter are available at: https://doi.org/10.6084/m9.figshare.12581678

*Author contributions.* WT wrote the manuscript with input from all authors and developed the OPTOkopter together with WH. UM initiated and supported the development and assisted with all measurements that are presented. All authors contributed to the discussion of the results.

*Competing interests.* WT, MH and UM developed the OPTOkopter while being employed at OPTOLUTION Messtechnik GmbH, a company that is aiming to commercialize measurement services with this drone.

*Acknowledgements.* We thank the Physikalisch-Technische Bundesanstalt, Arbeitsgruppe 1.41 Stroemungsmesstechnik / Gase and especially Michael Eggert for the opportunity to compare our wind speed measurements with the bistatic lidar.

We thank the Technische Universitaet Dresden, Fakultaet Maschinenwesen, Institut für Luft- und Raumfahrttechnik, Experimentelle Aerodynamik and especially Veit Hildebrand for the opportunity to fly inside the wind tunnel.

Thanks to Klaus-Peter Neitzke, (Hochschule Nordhausen) and Thomas Eipper (Technische Universitaet Dresden) for the assistance with measurements and photographs during the wind tunnel flights.

We thank the Eidgenössische Institut für Metrologie (METAS) for the opportunity to test the sonic anemometers in their wind tunnel.

Thanks to the Ökostrom Erzeugung Freiburg GmbH, Erwin Schlauderer for allowing us to measure the wind turbine wake.

Thanks to the Ardupilot community for developing a safe, great and open flight controller firmware.



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
