# Peer review of "Towards accurate and practical drone-based wind measurements with an ultrasonic anemometer"

_Atmospheric Measurement Techniques, 2020_

## Referee Comment (RC1) · Anonymous Referee #2 · 21 Nov 2020

The authors describe their first experiences with 3-D wind velocity measurements from a custom-designed quad-copter drone OPTOkopter, equipped with a special frame allowing for adoption of commercially available ultrasonic anemometers "TriSonica Mini Wind and Weather Sensor" by TriSonica and "Windmaster" by Gill Instruments. A description of the drone and it's design is provided, followed by a nicely documented review of the system properties retrieved from sophisticated wind tunnel and in-situ airborne tests where sensor readings were compared to velocity measurements from a bistatic Doppler lidar. Finally, example measurements performed in a wind turbine wake, documenting system capabilities are presented. The manuscript is written in a clear way and its content is of interest to the scientific community involved in boundary layer measurements and wind characterization. After addressing remarks pointed

below the manuscript should be published in AMT.

Remarks. l. 174 and Fig.4 It is not clear how data reported in Fig.4 were obtained. Add a comment, please.

Section 3.1. You show the dependence of accuracy of the sensor to pitch angle at 15 m/s windspeed only. Are the results at lower speeds (e.g. ∼5m/s) comparable? Does TriSonica perform relatively better?

Section 3.4. It seems that the flow in the course of the measurements was pretty turbulent. It would be interesting to compare the power spectral densities of velocities measured by the OPTOkopter and by the lidar . This would demonstrate capabilities of the system to measure and characterize atmospheric turbulence. Are any peaks related to the characteristic frequency related to stabilization of the drone are noticeable?

Editing suggestion: It could be useful to combine figs 10 and 11 19 into a two-panel figure and figs 14, 15, 16 and 18,19, 20 into three-panel figures?.

---

## Referee Comment (RC2) · Anonymous Referee #1 · 21 Nov 2020

The paper "Towards accurate and practical drone-based wind measurementswith an ultrasonic anemometer" by Thielicke et al describes a custom UAV platform set up for wind measurements. This platform has is a light (weight <5kg) quad-copter with relatively long flight time (endurance >45min). The platform equipped with either one of two ultrasonic anemometers (Trisonica and Windmaster) was tested in various enviroments (wind tunel, open air, turbulent flow behind a wind turbine) and validated with reference measurements (anemometers, wind lidars). Results show

The paper is very well organized as well as written. In particular, I applaud Authors for a concise, yet extensive and up-to-date review of UAV-based wind measurement attempts with discussion of techniques used and their findings. The discussion of results is well organized and narrative is clear. In my opinion paper's finding are well

supported by the data and analysis and I suggest minor revision of the paper.

Major Comments:

The key shortcoming, in my opinion, is that the paper is some parts if not achieving its full potential. The authors put a lot of effort to quantify performance of their system in turbulent environments. On the other hand, little attention is on relatively undisturbed flow within PBL. One solution would be to include comparison between UAV based measurements and wind lidar (Section 3.4) at altitude where flow is less disturbed by trees, buildings and other obstacles (100m?).

When considering wind field in general meteorological context, we tend to separate horizontal and vertical wind components, because they are differentiated by typical values as well are relevant processes. Typically vertical wind velocity is 1-2 orders of magnitude smaller than horizontal wind velocity. What is a bias when measuring in atti mode vs hoover? In my opinion, some discussion of errors in horizontal and vertical wind components, separately, would be of interest to the community.

My understanding is that Trisonica was tested in wind tunnel only. However, it would be interesting to see validation of the two sensors in realistic conditions, including turbulent and relatively undisturbed flows.

Authors claim that Optokopter is bettern than COST plaftorms because its endurance is longer. I understand that it may translate into statistics of measured variables, but the manuscript fails, in my opinion, to recognize that advantage. What do I need 45 min flight? Why not 20min? Some discussion of a measurement duration (e.g. how long do I need to measure at single 'point' to get the most accurate data) would be of interest.

Finally, what are lessons learned and how they translate into other platforms. One message I get is that Windmaster is better than Trisonica (only in wind tunnel, but that likely propagates into realistic flows). Assuming that I do not want to buy Optokopter

(or commission it for my measurements), how can I perform more accurate wind measurements with my platform of choice.

Minor Comments:

Sections 3.2, 3.3, - which anemometer was used? Trisonica or Windmaster? I'd expect the latter but I did not find this information in the text.

I'd suggest combining Figures 14, 15 and 16 into single figure with 3 panels.

I'd suggest combining Figures 18, 19 and 20 into single figure with 3 panels.

---

## Author Comment (AC1) · 18 Dec 2020

Anonymous Referee #1,

Dear Referee, thank you very much for taking the time to review our manuscript! We think that by incorporating your input into our manuscript (see details below), we have improved the quality and scientific significance of the paper. Here are our replies and the changes that we implemented:

1. **Little attention is paid to undisturbed flow in PBL. Include a comparison between wind lidar at higher altitudes.**
   a. Measurements in turbulent environment with a UAV appears to be most challenging to the authors, that is why we put a focus on this aspect. Data at higher altitudes (80 m and 100 m) is however included in Table 2. Flying at more than 100 m height would have required a special permit at the lidar measurement site. Additionally, the lidar data rate decreases and the measurement volume increases at altitudes > 100 m. We think that if our drone system is able to capture highly turbulent flow with small bias and RMSE, then this would only improve with altitude as the accuracy of our sonic anemometer does not depend on altitude.

2. **Discuss errors of vertical and horizontal wind component separately.**
   a. We agree that vertical and horizontal wind should be discussed separately. We implemented these changes:
      - Added Figure 10 (right) to section 3.2 showing the vertical wind component during wind tunnel testing of the full drone
      - Added the following text in section 3.2: The story is different for the wind elevation and the vertical speed. Here, the propeller induced flow has a large effect for wind speeds <= 10 m/s (see Figures 11 and 12). The wind elevation bias at very low wind speeds reaches 11°, and the vertical speed bias is around 0.4 m/s.
      - Added the column "vertical speed" in table 2, showing vertical speed bias and vertical speed RMSE in comparison the lidar measurement.
      - The conclusions do not change as the existing sentence "Propeller-induced flow mainly adds a vertical component to the flow without adding a horizontal component - even at large pitch angles. The vertical component can effectively be compensated by subtracting a value that is proportional to the mean motor throttle." does already include a suitable statement about the vertical flow component.

3. **Test the Trisonica also on the UAV, not only in wind tunnel**
   a. The initial plan was to use the Trisonica for UAV based wind measurements, so we acquired this device. When mounted on a drone, the sensor will be tilted because of the drone's pitch angle. Additionally, the propellers induce a vertical flow component which increases the resulting angle of attack at the sensor. During initial wind tunnel testing, we discovered that this sensor has problems to measure flows at high tilt angles. The error is not negligible and very difficult to compensate, because it also depends a lot on the yaw angle. This is why it was decided early in the design process that the accuracy will not be sufficient for the application. However, there are

already applications that use the Trisonica on a gimbal which ensures that the tilt angle is kept low. These applications will not be able to measure the vertical flow component, so measurements of the up- and downwash of a wind turbine (as presented in this paper) would not be possible. Therefore, this sensor is not suitable for our application, and consequently, we didn't test in in flight. We implemented the following change:

- Added the following sentence at the end of section 3.1: This study therefore focusses on in-flight measurements with the GILL Windmaster full-size sonic anemometer.

4. **Authors claim that the Optokopter is better than COTS platforms due to a longer flight time. Why is a long flight time advantageous? Describe advantage also in relation to statistical measures.**

a. We do not explicitly claim that the drone is better than COTS platforms, but we are happy that the referee perceived it in this way! There are several advantages of long endurance which we admittedly did not clearly state in the manuscript: A: Figure 15 shows the advantage of long flight time. Bias and RMSE decrease with averaging interval length. The longer the averaging interval, the less data points can be measured during a single flight. B: When measuring at high altitudes, a significant portion of the flight is spent to reach the measuring location. Having a shorter flight time would at some point make measurements at high altitudes unreasonable, because the drone would spent most of the endurance for climbing and descending after an exchange of the battery. C: Discharging a lipo battery at close to 1C (which would result in 1 hr flight time) is beneficial, as high-capacity lipos should not be charged at more than 2C. Charging at 2C results in a theoretical recharge time of 30 minutes, however due to the gradual reduction of charge current at the end of the charging process, a full charge typically takes 45 minutes. With the OPTOkopter, we can therefore fly almost uninterruptedly with only two lipo batteries. Flying with a lipo as large as possible ensures that the time between these interruptions is long (45 minutes in our case). The manuscript doesn't state these advantages; therefore, we implemented these changes:

- Updated Figure 4 with range
- Added the following in section 2.2: When measuring at remote locations and / or at high altitude, a significant portion of the flight time is spent for reaching the measurement location. Therefore, a long flight time is beneficial, as a larger fraction of the total endurance can be spent for acquiring wind speed at the desired location. This allows for longer averaging intervals and / or more measurement locations in a single flight. Significantly less time for swapping batteries is spent during long measurement campaigns. Additionally, the drone is equipped with a dual power supply. Batteries can be swapped without cutting power of the drone, so no reboot or GNSS reacquisition is required.

5. **Describe in more detail the lessons learned: How can an existing design be optimized for most accurate wind measurements?**

a. We have put more focus on the "lessons learned" in the update:

- Renamed Conclusions section to Conclusions and recommendations
- Added in conclusions: The performance of an anemometer that is to be installed on a drone should be verified at suitable tilt angles in a precision wind tunnel. The maximum tilt angle needs to be determined with drone test flights at the maximum desired wind speed.

- Added in conclusions: Placing the wind sensor far away from the rotors is a key requirement for this simple correction to work: As has been shown in Figure 6, flow distortion at the sonic anemometer is very small. This ensures that changing the pitch angle of the drone will not change the amount of flow distortion that is present at the anemometer location. In this case, a simple correction for the vertical flow component, that depends only on the average motor throttle, can be used. The gain of this correction should ideally be determined and verified in a large wind tunnel.
- Mounting an anemometer on such a long lever arm significantly increases the moment of inertia of the drone. It is therefore necessary to adjust the control loop parameters (e.g. proportional gain (P) of roll and pitch was increased by a factor of 4 and derivative gain (D) by a factor of 3). Care has to be taken to mount the anemometer exactly on top of the centre of gravity, otherwise roll and pitch motion of the drone results in an additional yaw moment. Yaw control is typically the weakest axis in quadrotors, so this could lead to serious control problems during flight. Furthermore, the anemometer mount needs to be very stiff in roll, pitch and yaw axes, otherwise oscillations (due to the increased P and D) are very likely to happen.

6. **Add information about which anemometer was used in Section 3.2 and 3.3,**
    - Added sentence in first paragraph of section 3.2: The GILL Windmaster was used as wind measuring device on the drone (see Figure 9).
    - Added sentence in first paragraph of section 3.3: After testing the performance of individual components in the measurement system, the accuracy of the full flying setup (OPTOkopter with GILL Windmaster and all compensations running) was assessed.

7. **Combine Figures 14-16**
    a. Done
8. **Combine figures 18-20**
    a. Done

**Additional changes to the manuscript due to an update in the contributing authors**

As the list of authors for this manuscript was updated, there are additional minor changes to the manuscript that we describe below:

1. Updated the wording so it is compatible with different parties being involved in the presented research. E.g. replaced ("We developed something" by "Something was developed").
2. The PTB lidar is validated at 8 meters height, but not at higher altitude. Therefore, using the word "validated" when presenting data at altitudes higher than 20 m is misleading. We replaced it with "analysed".
3. Updated all figures with correct axis labels (e.g. "Power [W]" replaced by "Power in W")
4. Updated Abstract: Removed "average absolute bias", as this is not comprehensible. Replaced this with a range of bias and RMSE for all the flights we performed.
5. Updated the Abstract with details on how an improved accuracy was reached:
   - "Key requirements for the accuracy are the use of a full-size sonic anemometer, a large distance between anemometer and propellers, and using a suitable algorithm for reducing the effect of propeller-induced flow."
6. Section 3.1: Instead of constantly jumping between Windmaster and Trisonica, each instrument now has its own paragraph, Trisonica is discussed first, therefore Table 1 shows the Trisonica on the left now.
7. Section 3.2: Mentioned more clearly that we used data recorded during free flight tests for setting parameters of pitch, and front / rear motor throttles in the wind tunnel:
   - "The drone was fixed to a rigid mount during most of the measurements, but using the data from normal free flight allowed to set realistic motor throttles and pitch angles for each wind tunnel speed."
8. Section 3.2: Results of the free flight inside the wind tunnel stated more clearly:
   - "This test confirmed that motor throttles and pitch angle during free wind tunnel flight and during free outside flight were in close agreement."
9. Re-worked the power spectral density plot (Figure 12 in the initial submission). Initially, we took wind speed magnitude (scalar quantity, calculated from all velocity components) for the analysis. A better approach is to directly use a velocity component. As we were oscillating in East-West direction, we took the corresponding velocity component. The resulting plot is more suitable and correct, and still shows that the motion of the drone is suppressed by a factor of about 13. The wording in all places that reference this result was adjusted accordingly.
10. Section 3.4: Removed "(within 1 m range)", as the exact (sub-second) synchronization between the timing unit in the lidar and in the drone has not been tested.
11. Section 3.4: Changes sentence about correlation of TI and RMSE to "A significant linear correlation was found also for turbulence intensity and wind speed RMSE (r = 0.76), azimuth RMSE (r = 0.90) and elevation RMSE (r = 0.87)."
12. Section 3.4: Replaced the "Conditions for high / low RMSE" equation with the sentence "A large distance between the measurement volumes, together with a high TI, will therefore result in high RMSE."
13. Section 3.4: Added sentence "The relative position of the OPTOkopter to the measurement volume of the lidar changed significantly while we were flying circles around the lidar. If there would be a significant influence from the OPTOkopter, then this should be visible as periodic bias error, but this is not the case."

14. Figure 16 (original Figure nr. 21), left: Did show slightly incorrect scaling, fixed
15. Conclusions: Replaced "Based on our tests of the individual components and the full system, we think that the total measurement accuracy is equal to the accuracy of the anemometer alone, as tested in the wind tunnel." with "Based on our tests of the individual components and the full system, we think that mounting the anemometer on our drone does not significantly increase the measurement uncertainty of the anemometer."
16. The PTB lidar data has been removed from the open data repository after a request by the PTB.

Text Compare
Produced: 04.12.2020 13:20:08

Mode:  Differences, Ignoring Unimportant
Left file: \Publikation\Latex_Submission_1\Thielicke_AMT_drone_ultrasonic_v2.tex
Right file: \Publikation\Latex_Submission_2\Thielicke_AMT_drone_ultrasonic_v4.tex

| Left | | Right |
|---|---|---|
| | -+ | \Author[2]{Michael}{Eggert}
 \Author[2]{Paul}{Wilhelm} |
| | -+ | \affil[2]{Physikalisch-Technische Bundesanstalt, Department 1.4 Gas Flow, 38116 Braunschweig, Germany} |
| Wind data collection in the atmospheric boundary layer benefits from short term wind speed measurements using unmanned aerial vehicles. Fixed and rotary wing devices with diverse anemometer technology have been used in the past to provide such data, but the accuracy still has the potential to be increased. We developed a light weight drone (weight including sensor $\leq$~5~\unit{kg}) with long flight endurance (>~45~\unit{min}) for carrying an industry standard precision sonic anemometer. Accuracy tests have been performed with the isolated anemometer at high tilt angles in a calibration wind tunnel, with the drone flying in a large wind tunnel, and with the full system flying at different heights next to a bistatic lidar reference. | <> | Wind data collection in the atmospheric boundary layer benefits from short term wind speed measurements using unmanned aerial vehicles. Fixed and rotary wing devices with diverse anemometer technology have been used in the past to provide such data, but the accuracy still has the potential to be increased. A light weight drone for carrying an industry standard precision sonic anemometer was developed. Accuracy tests have been performed with the isolated anemometer at high tilt angles in a calibration wind tunnel, with the drone flying in a large wind tunnel, and with the full system flying at different heights next to a bistatic lidar reference. |
| The propeller-induced flow deflects the air to some extent, but this effect is compensated effectively. Our data fusion shows no signs of crosstalk between ground speed and wind speed. When compared with the bistatic lidar in very turbulent conditions, with 10~seconds averaging interval and with the UAV constantly circling around the measurement volume of the lidar reference, wind speed measurements have an average absolute bias of 1.9\% (0.073~\unit{m\,s^{-1}}), wind elevation average absolute bias is 0.5\unit{^{\circ}}, and wind azimuth average absolute bias is 1.5\unit{^{\circ}}, indicating excellent accuracy under challenging and dynamic conditions. The system was finally flown in the wake of a wind turbine, successfully measuring the spatial velocity deficit distribution during forward flight, yielding results that are in very close agreement to lidar measurements and the theoretical distribution. We believe that the results presented in this paper can provide important information for designing flying systems for precise air speed measurements either for short duration at multiple locations (battery powered) or for long duration at a single location (power supplied via cable). UAVs that are able to accurately measure three-dimensional wind might be used as cost effective and flexible addition to measurement masts and lidar scans. | <> | The propeller-induced flow deflects the air to some extent, but this effect is compensated effectively. The data fusion shows a substantial reduction of crosstalk (factor 13) between ground speed and wind speed. When compared with the bistatic lidar in very turbulent conditions, with 10~seconds averaging interval and with the UAV constantly circling around the measurement volume of the lidar reference, wind speed measurements have a bias between -2.0\% and 4.2\% (RMSE: 4.3\% to 15.5\%), vertical wind speed bias is between -0.05~\unit{m\,s^{-1}} and 0.07~\unit{m\,s^{-1}} (RMSE: 0.15~\unit{m\,s^{-1}} to 0.4~\unit{m\,s^{-1}}), elevation bias is between -1\unit{^{\circ}} and 0.7\unit{^{\circ}} (RMSE: 1.2\unit{^{\circ}} to 6.3\unit{^{\circ}}), and azimuth bias is between -2.6\unit{^{\circ}} and 7.2\unit{^{\circ}} (RMSE: 2.6\unit{^{\circ}} to 8.0\unit{^{\circ}}). 
[revised manuscript text omitted]

\includegraphics[width=12cm]{figures/fusion.pdf}

A miniature sonic anemometer \citep[TriSonica Mini Wind and Weather Sensor,][]{Anemoment2020} and a full-size, sonic anemometer \citep[factory precalibrated Windmaster,][]{Gill2020} were tested at 0~-~360\unit{^{\circ}} yaw with 0\unit{^{\circ}}, 10\unit{^{\circ}}, 20\unit{^{\circ}}, 30\unit{^{\circ}} pitch angles (see Fig. \ref{fig:fig02} and \ref{fig:metas_tunnel}) in a traceable wind tunnel (400 $\cdot$ 260~mm cross-section, 400~mm length; accredited according to ISO/IEC 17025, Eidgenössisches Institut für Metrologie METAS, Switzerland) at wind speeds between 1~\unit{m\,s^{-1}} < v < 15~\unit{m\,s^{-1}}. The measurements were compared with a calibrated propeller anemometer (measurement uncertainty 2\%) at 20\unit{^{\circ}}C, 950~hPa, 47\% humidity. Both anemometers were mounted in the wind tunnel using the same attachments as in the drone, including global navigation satellite system (GNSS) / magnetometer and cable connections to assure that measurement conditions reflect the flying drone.

\includegraphics[width=12cm]{figures/metas_tests.pdf}

A suitable anemometer for application on UAVs should be able to accurately sense wind speed, azimuth and elevation. This should be possible for all pitch, roll and yaw angles that occur during a typical measurement flight of the UAV. In the proposed design, the maximum pitch angle of the UAV at the maximum air speed (20~\unit{m\,s^{-1}}) is 30\unit{^{\circ}}.

Wind speed measurements with the Trisonica are lower than the reference (see Table \ref{tab:tri_vs_gill}). Note that the results in Table \ref{tab:tri_vs_gill} show bias and RMSE for measurements at 15~\unit{m\,s^{-1}} and 0 - 360\unit{^{\circ}} yaw angle at four different pitch angles. At large pitch angles there is a strong bias (-17.3\% with a RMSE of 16.2\%). At zero pitch, there still is a bias of -6.3\% and a RMSE of 5.4\%. For the Trisonica, no relation between pitch angle and elevation could be determined. The Trisonica was tested in the wind tunnel in November 2018. After these results were reported to the manufacturer, a firmware update (ver 1.7.0, February 2019) addressed the issue of wind shadowing, potentially enhancing the accuracy at zero pitch angle. We had no opportunity to test this firmware in a wind tunnel yet. The issue at higher pitch angles will most likely remain, as we think it is impossible to accurately measure a vertical wind component with a small device with an inherently high blockage ratio. The latest firmware still needs to be tested in a wind tunnel for 0-360\unit{^{\circ}} and several pitch angles to check for improvements in accuracy. Based on these measurements, we think that the accuracy of 3D wind measurements with hard-mounted miniature sonic anemometers on UAVs is limited, and might explain the limited accuracy that several studies report for in-flight measurements with these kind of sensors (see Introduction). Miniature sensors should be mounted on a stabilizing gimbal and with the UAV flying at constant altitude to ensure that the vertical wind component is kept low.

[revised manuscript text omitted]
 8~\unit{m,s^{-1}}, the azimuth by about 100\unit{^{\circ}}, and the elevation by about 67\unit{^{\circ}} during this selected measurement flight (see Figures \ref{fig:time_resolved_ptb_magn}, \ref{fig:time_resolved_ptb_azimuth} and \ref{fig:time_resolved_ptb_elevation}). These numbers emphasize the importance of being capable to measure three-dimensional wind with a suitable anemometer. Despite the dynamic situation, the comparison with the PTB lidar reference shows an excellent agreement of the three dimensional wind speed (see Figures \ref{fig:time_resolved_ptb_magn}, \ref{fig:time_resolved_ptb_azimuth} and \ref{fig:time_resolved_ptb_elevation}). Note that these figures show measurements that were taken with 10~\unit{Hz} sampling rate.
\begin{figure}[!h]
        \includegraphics[width=8.3cm]{figures/vergleich_kopter_lidar_time_resolved_10Hz.eps}
        \caption{Wind speed measurement at 10~Hz, comparison of measurement from the PTB lidar with the OPTOkopter at 40~m height. Wind speed varies between 1.5 and 9.5~\unit{m\,s^{-1}}.}
        \label{fig:time_resolved_ptb_magn}
\end{figure}

\includegraphics[width=8.3cm]{figures/vergleich_kopter_lidar_time_resolved_azimuth_10Hz.eps}
        \caption{Wind azimuth measurement at 10~Hz, comparison of measurement from the PTB lidar with the OPTOkopter at 40~m height. Wind azimuth varies between 200 and 300\unit{^{\circ}}.}
        \label{fig:time_resolved_ptb_azimuth}
\end{figure}

\begin{figure}[!h]

\includegraphics[width=8.3cm]{figures/vergleich_kopter_lidar_time_resolved_elevation_10Hz.eps}
        \caption{Wind elevation measurement at 10~Hz, comparison of measurement from the PTB lidar with the OPTOkopter at 40~m height. Wind elevation varies between -42 and +25\unit{^{\circ}}.}

        \label{fig:time_resolved_ptb_elevation}

A linear Deming regression of the data in 1~\unit{s} averaging intervals has a slope of 1.03 and an offset of -0.03. The correlation coefficient is 0.95, indicating a good linear relation between both methods (see Figure \ref{fig:ptb_deming}).

We also analysed the effect of increasing averaging intervals (between 0.1 and 100~\unit{s}) on bias and RMSE. As expected, bias does hardly change when the averaging interval is increased. For the wind speed measurement, bias is between 2.9 and 3.7~\%, which is very similar to what we determined for the isolated Windmaster in the calibration wind tunnel (see Table \ref{tab:tri_vs_gill}). The wind speed RMSE is strongly dependent on averaging interval: It drops from 12\% at 0.1~\unit{s} to 1\% at 100~\unit{s} (see Figure \ref{fig:bias_rmse_vs_averaging_ptb_speed}).

The azimuth has a constant bias of about 2.6\unit{^{\circ}} and a RMSE decreasing from 7\unit{^{\circ}} to 1.9\unit{^{\circ}} (see Figure \ref{fig:bias_rmse_vs_averaging_ptb_azimuth}). The offset can result from a misalignment of the lidar, or interference of the compass on the UAV. We believe that this uncertainty is acceptable. It could be improved by fusing the heading measurements of the UAV with additional sensors like dual RTK GPS rovers. The elevation bias is constantly at about 0.4\unit{^{\circ}}. RMSE decreases from 7\unit{^{\circ}} at 0.1~\unit{s} averaging interval to 1\unit{^{\circ}} at 100~\unit{s} (see Figure \ref{fig:bias_rmse_vs_averaging_ptb_elevation}).

The bistatic PTB lidar and our OPTOkopter hence give closely matched results, even at 10~\unit{Hz} sampling interval. Naturally, this consistency increases with longer averaging intervals. When measuring at slightly different locations, the influence of spatial and temporal wind speed differences decreases with longer averaging intervals, lowering RMSE. Long averaging intervals also reduce measurement noise of both methods, again decreasing RMSE.

Turbulence intensity also shows a significant correlation with wind speed RMSE (r = 0.76), azimuth RMSE (r = 0.90) and elevation RMSE (r = 0.87). As a matter of course, TI also decreases when the averaging interval is increased, yielding lower RMSE (see Figures \ref{fig:bias_rmse_vs_averaging_ptb_speed}, \ref{fig:bias_rmse_vs_averaging_ptb_azimuth}, \ref{fig:bias_rmse_vs_averaging_ptb_elevation}). To conclude:
\begin{itemize}
        \item Conditions for high RMSE: ($\upuparrows\Delta t$ \textbf{AND} $\upuparrows TI$) \textbf{OR} $\downdownarrows m_{precision}$
        \item Conditions for low RMSE: ($\downdownarrows\Delta t$ \textbf{OR} $\downdownarrows TI$) \textbf{AND} $\upuparrows m_{precision}$
        \item[] with $\Delta t$ = time lag between measurement volumes = distance divided by mean wind speed, TI = turbulence intensity, $m_{precision}$ = measurement precision (inverse of

\caption{Amplitude spectrum of ground speed (as reported by the EKF2 in the flight controller), air speed (as reported by the sonic anemometer) and wind speed (as calculated by the data fusion) during a flight were the UAV was repeatedly oscillating in East-West direction between two waypoints. There is a clear peak at 0.104~\unit{Hz} in ground speed and air speed (which corresponds to the oscillation between waypoints), but a less distinctive peak in wind speed. This indicates, that the effects on wind speed measurements caused by translation and rotation of the UAV are suppressed by a factor of 13.4 by the data fusion.}

We compared the drone wind measurements with the bistatic Doppler lidar, developed at the Physikalisch-Technische Bundesanstalt (PTB) in Braunschweig, Germany \citep{Oertel2019,Mauder2020}. A data output rate of 10~\unit{Hz} was used in the PTB lidar, and different heights between 20 and 100~\unit{m} were tested.

After performing several flights, we selected a measurement at 40~\unit{m} height for a detailed analysis, as the correlation between wind speed measurements of the two methods was maximal for this dataset, indicating that we were flying very close to the measurement volume of the PTB lidar. We compared the data of the lidar reference to the drone measurement using an orthogonal Deming regression. RMSE and bias (based on paired observations) were determined for all measurement flights.

The OPTOkopter was always hovering at the lee side of the measurement volume. Wind speed, azimuth and elevation were sampled during multiple short flights of 10 minutes.
    Additionally, we were measuring wind speeds while circling (4~\unit{m} radius, 2.5~\unit{m\,s^{-1}} flight speed) around the lidar measurement volume to check for non-zero ground-speed related errors (see Fig. \ref{fig:figLidar}).

The measurement volume of the lidar at 40~\unit{m} height is surrounded by tall buildings and trees, generating highly unsteady flow: The wind speed varies by 8~\unit{m,s^{-1}}, the azimuth by about 100\unit{^{\circ}}, and the elevation by about 67\unit{^{\circ}} during this selected measurement flight (see Fig. \ref{fig:time_resolved_ptb_all}). These numbers emphasize the importance of being capable to measure three-dimensional wind with a suitable anemometer. Despite the dynamic situation, the comparison with the PTB lidar reference shows an excellent agreement of the three dimensional wind speed (see Fig. \ref{fig:time_resolved_ptb_all}). Note that these figures show measurements that were taken with 10~\unit{Hz} sampling rate.

\includegraphics[width=12cm]{figures/vergleich_kopter_lidar_time_resolved_10Hz_3panel.eps}
        \caption{Wind speed (left), azimuth (middle) and elevation (right) measurement at 10~Hz, comparison of measurement from the PTB lidar with the OPTOkopter at 40~m height. Wind speed varies between 1.5 and 9.5~\unit{m\,s^{-1}}. Wind azimuth varies between 200 and 300\unit{^{\circ}}. Wind elevation varies between -42 and +25\unit{^{\circ}}.}
        \label{fig:time_resolved_ptb_all}

A linear Deming regression of the data in 1~\unit{s} averaging intervals has a slope of 1.03 and an offset of -0.03. The correlation coefficient is 0.95, indicating a good linear relation between both methods (see Fig. \ref{fig:ptb_deming}).

We also analysed the effect of increasing averaging intervals (between 0.1 and 100~\unit{s}) on bias and RMSE. As expected, bias does hardly change when the averaging interval is increased. For the wind speed measurement, bias is between 2.9 and 3.7~\%, which is very similar to what was determined for the isolated Windmaster in the calibration wind tunnel (see Table \ref{tab:tri_vs_gill}). The wind speed RMSE is strongly dependent on averaging interval: It drops from 12\% at 0.1~\unit{s} to 1\% at 100~\unit{s} (see Fig. \ref{fig:bias_rmse_vs_averaging_ptb_all}, left).

The azimuth has a constant bias of about 2.6\unit{^{\circ}} and a RMSE decreasing from 7\unit{^{\circ}} to 1.9\unit{^{\circ}} (see Fig. \ref{fig:bias_rmse_vs_averaging_ptb_all}, middle). The offset can result from a misalignment of the lidar, or interference of the compass on the UAV. We believe that this uncertainty is acceptable. It could be improved by fusing the heading measurements of the UAV with additional sensors like dual RTK GPS rovers. The elevation bias is constantly at about 0.4\unit{^{\circ}}. RMSE decreases from 7\unit{^{\circ}} at 0.1~\unit{s} averaging interval to 1\unit{^{\circ}} at 100~\unit{s} (see Fig. \ref{fig:bias_rmse_vs_averaging_ptb_all}, right).

Bistatic lidar and OPTOkopter hence give closely matched results, even at 10~\unit{Hz} sampling interval. Naturally, this consistency increases with longer averaging intervals. When measuring at slightly different locations, the influence of spatial and temporal wind speed differences decreases with longer averaging intervals, lowering RMSE. Long averaging intervals also reduce measurement noise of both methods, again decreasing RMSE.

A significant linear correlation was found also for turbulence intensity and wind speed RMSE (r = 0.76), azimuth RMSE (r = 0.90) and elevation RMSE (r = 0.87). As a matter of course, the measured TI also decreases when the averaging interval is increased, yielding lower RMSE (see Fig. \ref{fig:bias_rmse_vs_averaging_ptb_all}). A large distance between the measurement volumes, together with a high TI, will therefore result in high RMSE.

**Left version:**

random error).
\end{itemize}

Despite sometimes we were flying very close to the lidar, we believe that the presence of the UAV did not significantly change the flow in the measurement volume of the lidar: The measurements of the OPTOkopter have been successfully compensated for propeller induced flow (see Figure \ref{fig:wind_tunnel_speed} and \ref{fig:wind_tunnel_vertical}). If the OPTOkopter would have changed e.g. the vertical flow component in the measurement volume of the lidar, then there would be a large discrepancy between (compensated) OPTOkopter measurement and (uncompensated) lidar measurement. Furthermore, measurements at more remote locations that cannot influence the measurement of the lidar due to spatial separation (see Table \ref{tab:bias_rmse_ptb_all} for bias and RMSE of all tests we performed) have a very similar bias and RMSE.

\begin{figure}[!h]
    \includegraphics[width=8.3cm]{figures/bias_RMSE_vs_averaging_magn.eps}
    \caption{Bias and RMSE of the UAV wind speed measurement at 40~\unit{m} height for averaging intervals between 0.1~\unit{s} (10~\unit{Hz}) and 100~\unit{s} (0.01~\unit{Hz}). The PTB lidar is used as reference instrument.}
    \label{fig:bias_rmse_vs_averaging_ptb_speed}
\end{figure}

    \includegraphics[width=8.3cm]{figures/bias_RMSE_vs_averaging_azimuth.eps}
    \caption{Bias and RMSE of the UAV azimuth measurement at 40~\unit{m} height for averaging intervals between 0.1~\unit{s} (10~\unit{Hz}) and 100~\unit{s} (0.01~\unit{Hz}). The PTB lidar is used as reference instrument.}
    \label{fig:bias_rmse_vs_averaging_ptb_azimuth}
\end{figure}

\begin{figure}[!h]
    \includegraphics[width=8.3cm]{figures/bias_RMSE_vs_averaging_elevation.eps}
    \caption{Bias and RMSE of the UAV wind elevation measurement at 40~\unit{m} height for averaging intervals between 0.1~\unit{s} (10~\unit{Hz}) and 100~\unit{s} (0.01~\unit{Hz}). The PTB lidar is used as reference instrument.}

    \label{fig:bias_rmse_vs_averaging_ptb_elevation}

    \caption{Bias and RMSE of the OPTOkopter wind measurements at 10~s averaging interval, with the PTB lidar reference. The table includes data from all flights that were done. The distance to the measurement volume of the lidar was difficult to assess, but it was smaller than 10~\unit{m} in all cases. The comparison was done with the OPTOkopter hovering on spot or circling around the lidar measurement volume. Wind speed bias is generally low. RMSEs seem to increase with turbulence intensity. Average absolute bias of all the measurements: wind speed 2.0\%, elevation 0.4\unit{^{\circ}}, azimuth 3.3\unit{^{\circ}}. Average absolute bias of hovering flights: wind speed 2.1\%, elevation 0.4\unit{^{\circ}}, azimuth 4.7\unit{^{\circ}}. Average absolute bias of circling flights: wind speed 1.9\%, elevation 0.5\unit{^{\circ}}, azimuth 1.5\unit{^{\circ}}.}

The wind turbine (Enercon E 70 - E 4) is located in the Black Forest in southern Germany (47°45'53.43"N; 7°51'11.68"E) at about 1012~\unit{m} above sea level. The nacelle height is 85~\unit{m} and the rotor diameter (D) is 71~\unit{m} (see Figure \ref{fig:wind_turbine_situation}). Wind velocity was determined at 2~D behind the rotor disk. Flight duration was 22-minutes, and measurements were taken at 16~\unit{Hz}. The OPTOkopter was oscillating at constant altitude at nacelle height with a velocity of 5~\unit{m\,s^{-1}} on a path parallel to the rotor disk (see Figure \ref{fig:wind_turbine_situation} and \ref{fig:turbine_flight}). Because the wind speed was quite substantially varying with time (see Figure \ref{fig:wind_turbine_reference}), all wind measurements were normalized with the reference anemometer velocity on top of the nacelle ($u_{ref}$). Measurements were discretized in intervals of 1~\unit{m} along the flight path. Data from each of these bins was averaged.

A relatively constant velocity deficit ($\overline{u}/u_{ref}$) of 25\% is found behind the full diameter of the rotor disk. Further away from the rotor tips, the velocity becomes even larger than $u_{ref}$ (see Figure \ref{fig:turbine_wake}). Most likely, the reference anemometer is measuring velocities lower than the true free stream velocity, due to the proximity to the nacelle, and possible shadowing effects by the rotor blades.
When a wind turbine rotates clockwise (as viewed from the front), it will generate a swirl with anti-clockwise rotation. In a horizontal cross-section at nacelle height, this will result in air travelling down on the left side (again viewed from the front), and air travelling up on the right side. We captured the swirl (see Figure \ref{fig:wind_turbine_wake}, right), the magnitude is about 0.35~\unit{m\,s^{-1}} which is about 7.7\% of the average free stream velocity. The downwash is not perfectly symmetric around the centre of the wind turbine and may be influenced by the slope behind the wind turbine (see Figure \ref{fig:wind_turbine_situation}, middle).

We did not capture data at $>$ 1 z/D, as the wind direction slightly changed after the waypoints were positioned and uploaded to the UAV.
Our measurements are strikingly similar to theoretical velocity distributions \citep[e.g.][]{Wu2012,Keane2016} and lidar measurements in the wake of wind turbines \citep[e.g.][]{Vollmer2017,Menke2018}. We believe that the noise in the measurements is mostly due to the inconsistent free stream velocity (see Figure \ref{fig:wind_turbine_reference}) and that it can be decreased significantly by measuring for a longer duration (e.g. using more than one battery pack).

    \includegraphics[width=8.2cm]{figures/wind_turbine_flight.pdf}

    \includegraphics[width=12cm]{figures/wind_turbine_wake.pdf}

    \conclusions  %% \conclusions[modified heading if necessary]

The environmental science of the atmospheric boundary layer benefits from wind speed measurements collected by UAVs. We designed a suitable light weight rotary wing UAV for carrying an anemometer. Drones can measure close to structures and they can be validated comfortably by hovering close to a reference instrument. Flight time is often an issue with UAV based measurements. In our design, the battery is responsible for 49\% of the total weight. It can be replaced by COTS power-tethering devices, that allow for much longer, uninterrupted measurement flights at a single location at different altitudes up to 100~m.

    The OPTOkopter uses a full-size, industry standard anemometer instead of a miniature version, as the accuracy in three-dimensional flow is by a magnitude better. Measurements at the test site of the PTB lidar have shown that three-dimensional flow is highly likely to happen in situ, even when the OPTOkopter hovers on spot at a constant altitude. Due to the high contribution of vertical flow, using a single miniature sonic anemometer does not seem to be

**Right version:**

Despite sometimes we were flying very close to the lidar, we believe that the presence of the UAV did not significantly change the flow in the measurement volume of the lidar: The measurements of the OPTOkopter have been successfully compensated for propeller induced flow (see Fig. \ref{fig:wind_tunnel_all}). If the OPTOkopter would have changed e.g. the vertical flow component in the measurement volume of the lidar, then there would be a large discrepancy between (compensated) OPTOkopter measurement and (uncompensated) lidar measurement. The relative position of the OPTOkopter to the measurement volume of the lidar changed significantly while we were flying circles around the lidar. If there would be a significant influence from the OPTOkopter, then this should be visible as periodic bias error, but this has not been observed in the data.

    \includegraphics[width=12cm]{figures/bias_RMSE_vs_averaging_3panels.eps}
    \caption{Bias and RMSE of the UAV wind speed (left), azimuth (middle) and elevation (right) measurement at 40~\unit{m} height for averaging intervals between 0.1~\unit{s} (10~\unit{Hz}) and 100~\unit{s} (0.01~\unit{Hz}). The PTB lidar is used as reference instrument.}
    \label{fig:bias_rmse_vs_averaging_ptb_all}

    \caption{Bias and RMSE of the OPTOkopter wind measurements at 10~s averaging interval, with the PTB lidar reference. The table includes data from all flights that were done. The distance to the measurement volume of the lidar was difficult to assess, but it was smaller than 10~\unit{m} in all cases. The comparison was done with the OPTOkopter hovering on spot or circling around the lidar measurement volume. Wind speed bias is generally low. RMSEs seem to increase with turbulence intensity.}

The wind turbine (Enercon E 70 - E 4) is located in the Black Forest in southern Germany (47°45'53.43"N; 007°51'11.68"E) at about 1012~\unit{m} above sea level. The nacelle height is 85~\unit{m} and the rotor diameter (D) is 71~\unit{m} (see Fig. \ref{fig:wind_turbine_situation}). Wind velocity was determined at 2~D behind the rotor disk. Flight duration was 22-minutes, and measurements were taken at 16~\unit{Hz}. The OPTOkopter was oscillating at constant altitude at nacelle height with a velocity of 5~\unit{m\,s^{-1}} on a path parallel to the rotor disk (see Fig. \ref{fig:wind_turbine_situation} and \ref{fig:turbine_flight}). Because the wind speed was quite substantially varying with time (see Fig. \ref{fig:wind_turbine_reference}), all wind measurements were normalized with the reference anemometer velocity on top of the nacelle ($u_{ref}$). Measurements were discretized in intervals of 1~\unit{m} along the flight path. Data from each of these bins was averaged.

A relatively constant velocity deficit ($\overline{u}/u_{ref}$) of 25\% is found behind the full diameter of the rotor disk. Further away from the rotor tips, the velocity becomes even larger than $u_{ref}$ (see Fig. \ref{fig:turbine_wake}). Most likely, the reference anemometer is measuring velocities lower than the true free stream velocity, due to the proximity to the nacelle, and possible shadowing effects by the rotor blades.
When a wind turbine rotates clockwise (as viewed from the front), it will generate a swirl with anti-clockwise rotation. In a horizontal cross-section at nacelle height, this will result in air travelling down on the left side (again viewed from the front), and air travelling up on the right side. The swirl was captured (see Fig. \ref{fig:turbine_wake}, right), the magnitude is about 0.35~\unit{m\,s^{-1}} which is about 7.7\% of the average free stream velocity. The downwash is not perfectly symmetric around the centre of the wind turbine and may be influenced by the slope behind the wind turbine (see Fig. \ref{fig:wind_turbine_situation}, middle).

Data at $>$ 1 z/D was not captured, as the wind direction slightly changed after the waypoints were positioned and uploaded to the UAV.
The results of the measurements are strikingly similar to theoretical velocity distributions \citep[e.g.][]{Wu2012,Keane2016} and lidar measurements in the wake of wind turbines \citep[e.g.][]{Vollmer2017,Menke2018}. The noise in the measurements most likely results from the inconsistent free stream velocity (see Figure \ref{fig:wind_turbine_reference}) and can presumably be decreased significantly by measuring for a longer duration (e.g. using more than one battery pack).

    \includegraphics[width=8.3cm]{figures/wind_turbine_flight.pdf}

    \includegraphics[width=12cm]{figures/wind_turbine_wake.eps}

[revised manuscript text omitted]

\href{https://doi.org/10.6084/m9.figshare.12581678}{https://doi.org/10.6084/m9.figshare.12581678}}
%% use this section when having only data sets available

\authorcontribution{WT wrote the manuscript with input from all authors and developed the OPTOkopter together with WH. UM initiated and supported the development and assisted with all measurements that are presented. All authors contributed to the discussion of the results.}
%% this section is mandatory

%\disclaimer{} %% optional section

We thank the Physikalisch-Technische Bundesanstalt, Arbeitsgruppe 1.41 Stroemungsmesstechnik / Gase and especially Michael Eggert for the opportunity to compare our wind speed measurements with the bistatic lidar.

We thank the Technische Universitaet Dresden, Fakultaet Maschinenwesen, Institut für Luft- und Raumfahrttechnik, Experimentelle Aerodynamik and especially Veit Hildebrand for the opportunity to fly inside the wind tunnel.

Thanks to Klaus-Peter Neitzke, (Hochschule Nordhausen) and Thomas Eipper (Technische Universitaet Dresden) for the assistance with measurements and photographs during the wind tunnel flights.

We thank the Eidgenössische Institut für Metrologie (METAS) for the opportunity to test the sonic anemometers in their wind tunnel.

[revised manuscript text omitted]

\dataavailability{Data of all measurements presented in this paper (except for the PTB lidar data) and additional information on the OPTOkopter are available at:
\href{https://doi.org/10.6084/m9.figshare.12581678}{https://doi.org/10.6084/m9.figshare.12581678}}
%% use this section when having only data sets available

\authorcontribution{WT wrote the manuscript with input from all authors, developed and operated the OPTOkopter together with WH. UM initiated and supported the development and assisted with all measurements that are presented. ME constructed the PTB lidar system and its signal processing. PW and ME operated the Doppler lidar and preprocessed its 10 Hz and 1 Hz raw data. All authors contributed to the discussion of the results.} %% this section is mandatory

We thank the Technische Universitaet Dresden, Fakultaet Maschinenwesen, Institut für Luft- und Raumfahrttechnik, Experimentelle Aerodynamik and especially Veit Hildebrand for the opportunity to fly inside the wind tunnel. Thanks to Klaus-Peter Neitzke, (Hochschule Nordhausen) and Thomas Eipper (Technische Universitaet Dresden) for the assistance with measurements and photographs during the wind tunnel flights. We thank the Eidgenössische Institut für Metrologie (METAS) for the opportunity to test the sonic anemometers in their wind tunnel. Thanks to the Ökostrom Erzeugung Freiburg GmbH, Erwin Schlauderer for allowing us to measure the wind turbine wake. Thanks to the Ardupilot community for developing a safe, great and open flight controller firmware.